# Intramuscular Neurotrophin-3 normalizes low threshold spinal reflexes, reduces spasms and improves mobility after bilateral corticospinal tract injury in rats

Claudia Kathe[1]*, Thomas Haynes Hutson[2], Stephen Brendan McMahon[1], Lawrence David Falcon Moon[1]*

[1]Neurorestoration Department, Wolfson Centre for Age-Related Diseases, King's College London, University of London, London, United Kingdom; [2]Division of Brain Sciences, Department of Medicine, Imperial College London, London, United Kingdom

**Abstract** Brain and spinal injury reduce mobility and often impair sensorimotor processing in the spinal cord leading to spasticity. Here, we establish that complete transection of corticospinal pathways in the pyramids impairs locomotion and leads to increased spasms and excessive mono- and polysynaptic low threshold spinal reflexes in rats. Treatment of affected forelimb muscles with an adeno-associated viral vector (AAV) encoding human Neurotrophin-3 at a clinically-feasible time-point after injury reduced spasticity. Neurotrophin-3 normalized the short latency Hoffmann reflex to a treated hand muscle as well as low threshold polysynaptic spinal reflexes involving afferents from other treated muscles. Neurotrophin-3 also enhanced locomotor recovery. Furthermore, the balance of inhibitory and excitatory boutons in the spinal cord and the level of an ion co-transporter in motor neuron membranes required for normal reflexes were normalized. Our findings pave the way for Neurotrophin-3 as a therapy that treats the underlying causes of spasticity and not only its symptoms.

**\*For correspondence:** claudia. kathe@epfl.ch (CK); lawrence. moon@kcl.ac.uk (LDFM)

**Competing interests:** The authors declare that no competing interests exist.

## Introduction

Up to 78% of patients with spinal cord injury develop spasticity within the first year (*Adams and Hicks, 2005*). Spasticity is defined as a velocity-dependent increase in tonic stretch reflexes with exaggerated tendon jerks, resulting from hyperexcitability of the stretch reflex, and intermittent or sustained involuntary activation of muscles (*Pandyan et al., 2005*). Current treatments like tizanidine, botulinum toxin and baclofen are only symptomatic, transiently effective and have varying side effects (*Adams and Hicks, 2005*; *Thibaut et al., 2013*). These symptomatic treatments, which aim at exaggerated reflexes and muscle tone, rarely improve other functional disabilities associated with upper motor neuron lesions such as overground locomotion (*Dietz and Sinkjaer, 2007*). There has been progress in understanding the potential underlying causes of spasticity. However, it remains challenging to model spastic behaviours in rodents. Models are needed to identify new candidate therapies for spasticity. Upper arm spasticity after spinal cord injury and stroke often persists and hampers rehabilitation and effective new therapies are badly needed (*Dietz and Fouad, 2014*; *Adams and Hicks, 2005*).

To date, spastic behaviours that have been examined in rat models include tail spasms and clonus (*Bennett et al., 2004*, *1999*) and spontaneous hindlimb and tail spasms during swimming (*Gonzenbach et al., 2010*). One main goal of our study was to model a variety of signs of forelimb

**eLife digest** Injuries to the brain and spinal cord cause disability in millions of people worldwide. Physical rehabilitation can restore some muscle control and improve mobility in affected individuals. However, no current treatments provide long-term relief from the unwanted muscle contractions and spasms that affect as many as 78% of people with a spinal cord injury. These spasms can seriously hamper a person's ability to carry out day-to-day tasks and get around independently. A few treatments can help in the short term but have side effects; indeed while Botox injections are used to paralyse the muscle, these also reduce the chances of useful improvements. As such, better therapies for muscle spasms are needed; especially ones that reduce spasms in the arms.

Rats with injuries to the spinal cord between their middle to lower back typically develop spasms in their legs or tail, and rat models have helped scientists begin to understand why these involuntary movements occur. Now, Kathe et al. report that cutting one specific pathway that connects the brain to the spinal cord in anesthetised rats leads to the development of spasms in the forelimbs as well. Several months after the surgery, the rats had spontaneous muscle contractions in their forelimbs and walked abnormally. Further experiments showed that some other neural pathways in the rats became incorrectly wired and hyperactive and that this resulted in the abnormal movements.

Next, Kathe et al. asked whether using gene therapy to deliver a protein that is required for neural circuits to form between muscles and the spinal cord (called neurotrophin-3) would stop the involuntary movements in the forelimbs. Delivering the gene therapy directly into the forelimb muscles of the disabled rats a day after their injury increased the levels of neurotrophin-3 in these muscles. Rats that received this treatment had fewer spasms and walked better than those that did not. Further experiments confirmed that this was because the rats' previously hyperactive and abnormally wired neural circuits became more normal after the treatment.

Together these results suggest that neurotrophin-3 might be a useful treatment for muscle spasms in people with spinal injury. There have already been preliminary studies in people showing that treatment with neurotrophin-3 is safe and well tolerated. Future studies are needed to confirm that it could be useful in humans.

spasticity in awake, freely moving rats. These signs include clonus (repeated muscle jerks), prolonged spasms, twitches (fast involuntary contractions) and hyperreflexia, which reflect all symptoms associated with human spasticity after upper motor neuron lesions.

We sought to understand the mechanisms underlying these functional sensorimotor abnormalities. Sensory inputs from muscles spindles to the spinal cord play a major role in regulating spinal motor circuitry organization and output, especially after CNS injury (*Takeoka et al., 2014*; *Akay et al., 2014*). Group I and II proprioceptive afferents innervating muscle spindles project via the dorsal root ganglia (DRG) into the spinal cord to synapse onto motor neurons or interneurons whilst Ib afferents from Golgi tendon organs form synapses on interneurons only (*Eccles et al., 1957*; *Dietz, 2002*). The spinal circuitry integrating these sensory signals has a highly specific patterning in order to produce complex muscle activation synergies (*Kiehn, 2016*). Proprioceptive neurons provide positive feedback about muscle contractions, which adjusts movements to environmental factors and controls reflex responses via monosynaptic and polysynaptic spinal pathways (*Dietz, 2002*).

After traumatic brain or spinal cord injury, the circuits below the lesion site lose some supraspinal input, but they can reorganize and adapt. This reorganization can involve maladaptation causing functional abnormalities such as spasticity. Proposed mechanisms of spasticity include increased activity and connectivity between proprioceptive Ia muscle afferents and motor neurons, reduced presynaptic inhibition of Ia afferents by spinal interneurons (*Toda et al., 2014*; *Kakinohana et al., 2012*), reduced autogenic inhibition from Golgi tendon organs (via Ib afferents and inhibitory interneurons), reduced reciprocal inhibition by Ia afferents from antagonist muscles and modified excitation and inhibition from muscle spindle group II afferents reviewed in [*Nielsen et al., 2007*;

*Dietz, 2002*]) as well as intrinsic changes in motor neurons such as altered ion channels, serotonergic receptors and transporter concentrations in the membranes (*Boulenguez et al., 2010*; *Murray et al., 2011*) and an increase in persistent inward currents in motor neurons (*Hultborn et al., 2013*; *Boulenguez et al., 2010*; *Bennett et al., 2001*; *ElBasiouny et al., 2010*) which together cause an increase in motor neuron excitability.

Peripherally derived Neurotrophin-3, a nerve growth factor, is required for survival of large diameter sensory neurons and motor neurons (*Ernfors et al., 1994*; *Woolley et al., 2005*) and for correct patterning of central proprioceptive afferents during development (*Patel et al., 2003*). Neurotrophin-3 can also reduce motor neuron excitability (*Petruska et al., 2010*). For these reasons, we elected to evaluate Neurotrophin-3 as a novel therapy for forelimb spasticity in adult rats after upper motor neuron lesions.

After ensuring the survival and patterning of sensory and motor circuitry during development, Neurotrophin-3 levels drop in the muscle (*Murase et al., 1994*). Interestingly, Neurotrophin-3 levels are again increased after exercise (*Gómez-Pinilla et al., 2001*; *Ying et al., 2003*), such as after rehabilitative training post-stroke or spinal cord injury (*Hutchinson et al., 2004*; *Côté et al., 2011*). Rehabilitative training which is based on the principle that coordinated afferent input and motor output improves functional recovery (*Frigon and Rossignol, 2006*; *Rossignol et al., 2006*; *Smith et al., 2006*) also normalises proprioceptive reflexes in spinal cord injury models (*Côté et al., 2011*). Input from muscle afferents is important for maintaining normal locomotion in adult mice (*Akay et al., 2014*) and for the reorganization of supraspinal pathways after spinal cord injury (because genetic deletion of muscle spindles leads to poorer outcomes [*Takeoka et al., 2014*]). Neurotrophin-3 is synthesized by muscle spindles in the periphery (*Copray and Brouwer, 1994*). Therefore, Neurotrophin-3 may serve as a muscle-derived signal that enhances functional recovery and neuroplasticity after spinal cord injury and peripheral supplementation of Neurotrophin-3 might enhance recovery.

Recent studies from our group show that peripherally administered Neurotrophin-3 can remodel spared corticospinal tract connections and promote sensorimotor recovery in adult and elderly rats after stroke (*Duricki et al., 2016*). Peripheral treatment with recombinant Neurotrophin-3 has already been tested in Phase I and II clinical trials and was found to be safe and well-tolerated (*Parkman et al., 2003*; *Chaudhry et al., 2000*; *Coulie et al., 2000*; *Sahenk, 2007*; *Sahenk et al., 2005*). Interestingly, Neurotrophin-3 also improved sensory and reflex function in patients with Charcot Marie Tooth 1A disease, a large diameter fiber sensorimotor neuropathy (*Sahenk et al., 2005*). Taken together, Neurotrophin-3 organizes supraspinal motor tracts and improves sensory function, but whether spinal sensorimotor circuitry and reflex changes can be regulated by Neurotrophin-3 in neuromotor disorders has not been demonstrated yet.

Here, we established a novel rodent model of spasticity which displays many aspects of functional spasticity observed in humans. The model involves bilateral corticospinal tract lesioning in the pyramids, which was sufficient to elicit changes in the connectivity of proprioceptive afferents to motor neurons resulting in signs of spasticity including hyperreflexia of mono- and polysynaptic reflexes. Intramuscular overexpression of Neurotrophin-3 normalized several spinal reflexes. Analysis of neurophysiological properties of the spinal circuitry and of molecular markers revealed that Neurotrophin-3 acts as a regulator of afferent input connectivity to spinal motor circuitry and as a regulator of motor neuron excitability. Taken together, our study established that proprioceptive afferent input strength and connectivity to the spinal cord changes after supraspinal injury, which affected motor output resulting in signs of spasticity and reduced functional recovery. Overexpression of Neurotrophin-3 in the muscles regulated afferent input specificity and re-balanced excitatory and inhibitory networks in the spinal cord improving functional and neurophysiological outcomes.

## Results

### Rats developed spasticity after bilateral pyramidotomy

To establish a rodent model of spasticity of the forelimb, we performed a bilateral transection of the pyramids in the brainstem (bPYX) of anesthetized rats, which interrupts the corticospinal tracts (*Figure 1A–C*; *Figure 1—figure supplement 1*; *Figure 2—figure supplement 1*). The lesion completeness was assessed for all rats at the end of the study with eriochrome cyanine staining of the

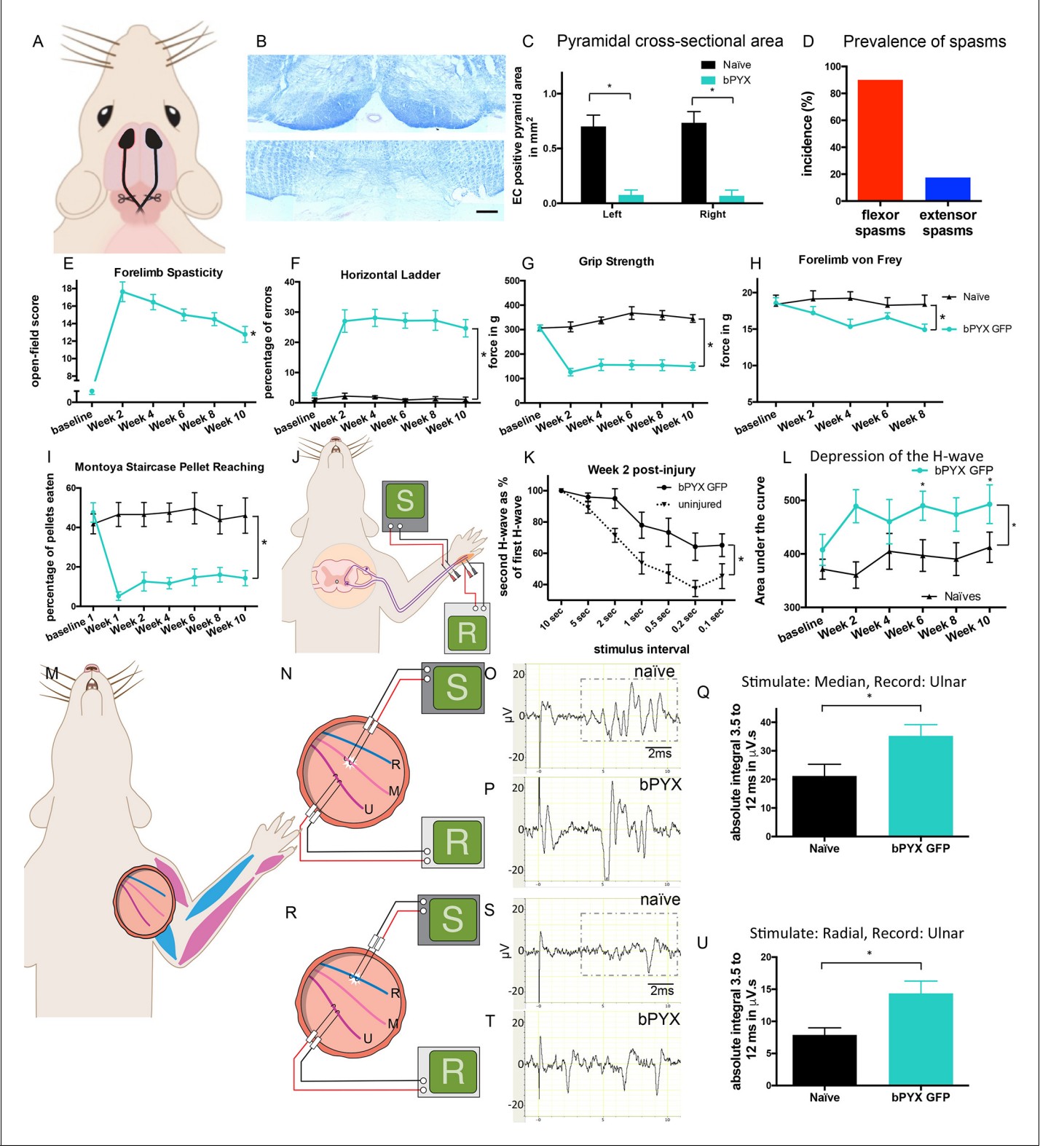

**Figure 1.** Bilateral transection of the corticospinal tracts in the pyramids resulted in forelimb spasms, impaired walking and caused hyperreflexia. (A) Schematic of experimental set-up. Rats received a bilateral pyramidotomy (bPYX) and were treated 24 hr post-injury with AAV1-NT3 or AAV1-GFP injections into *biceps brachii* and distal forelimb flexors and hand muscles. For clarity in describing the consequences of bPYX, *Figure 1* shows only data from uninjured naïve rats and from bPYX rats treated with AAV1-GFP (bPYX GFP). (B) Transverse sections of the medullary brainstem were stained with eriochrome cyanine to quantify the cross-sectional area of the pyramids in (upper panel) uninjured naïve rats and (lower panel) bPYX GFP rats.
*Figure 1 continued on next page*

*Figure 1 continued*

Pyramids are absent in the lower panel. Scale bar: 200 µm. (C) The left and right pyramids were almost completely absent in bPYX GFP rats versus uninjured naïve rats (RM two-way ANOVA, group F = 107.4, p<0.0001; bPYX GFP versus naïve p<0.001). (D) A greater percentage of bPYX GFP rats exhibited flexor spasms than the percentage that exhibited extensor spasms two weeks post-injury. (E) Rats showed abnormal forelimb movements and signs of spasticity in the open field after bPYX (RM two-way ANOVA, group F = 19.8, p<0.001; bPYX GFP baseline versus all post-injury Weeks, paired t-test p-values<0.05). (F) Bilateral pyramidotomy caused rats to make many errors on the horizontal ladder with their treated forelimb as a percentage of the total steps taken (RM two-way ANOVA, group F = 123.4, p<0.001; bPYX GFP vs naïve p<0.0001; bPYX GFP vs naïve at Weeks 2, 4, 6, 8 and 10, p-values<0.05). (G) Unilateral Grip Strength Test. bPYX rats had reduced grip strength with their treated forepaw at Weeks 2, 4, 6, 8 and 10 versus uninjured naïves (RM two-way ANOVA, group F = 145.0, p<0.001; bPYX GFP vs naïve p<0.0001; bPYX GFP vs naïve at Weeks 2, 4, 6, 8 and 10, p-values<0.05). (H) Responses to mechanical stimulation of the treated forepaw were assessed using the automated von Frey test. Bilateral pyramidotomy caused slight mechanical hypersensitivity (RM two-way ANOVA, group F = 5.2, p=0.019; bPYX GFP vs naïve p = 0.003; bPYX GFP versus naïve at Week 4 and 8, p-values<0.05) (I) Montoya Staircase Pellet Reaching Test. This test assesses fine motor function of the distal forelimb. The sucrose pellets that were eaten on the treated side were counted. Bilateral pyramidotomy led to a persistent deficit in dexterity (RM two-way ANOVA, group F = 100.3, p<0.0001; bPYX GFP vs naïve p<0.0001; bPYX GFP versus naïve at weeks 1, 2, 4, 6, 8 and 10 post-injury p-values<0.0001). (J) Schematic showing the H-reflex paradigm. The ulnar nerve was stimulated distally and EMGs were recorded from a homonymous hand muscle (*abductor digiti quinti*). (K) Frequency-dependent depression: The H-wave was depressed at short inter-stimulus intervals in uninjured naïve rats. Less H-wave depression was observed in bPYX GFP rats two weeks post injury (RM two-way ANOVA, group F = 9.8, p<0.0001; uninjured versus bPYX GFP at 2 s, 1 s, 0.5 s, 0.2 s and 0.1 s inter-stimulus interval p-values<0.05). (L) Data for each rat at each week was analysed by measuring the area under each curve which gives an electrophysiological correlate of hyperreflexia. Bilateral pyramidotomy caused an increase in hyperreflexia relative to uninjured naïve rats at week 2, 4, 6, 8 and 10 (RM two-way ANOVA, group F = 5.9, p<0.001; bPYX GFP versus naïve p = 0.003; bPYX GFP versus naïve at weeks 2, 6 and 10 post-injury, p-values<0.05). (M) At week 10, the radial, median and ulnar nerves were exposed for stimulation and recording. The radial nerve (blue) innervates extensor muscles (blue) whereas the median and ulnar nerves (pink and magenta) innervate synergist flexor muscles (magenta). (N) Stimulation of afferents in the median nerve evoked responses in the (synergist, flexor) ulnar nerve. (O–P) Example traces show recordings from (O) uninjured naïve and (P) bPYX GFP rats. The boxed area highlights polysynaptic compound action potentials which were analysed for *Figure 1Q*. (Q) The polysynaptic compound action potentials were quantified by measuring the absolute integral (area under the rectified curve) from 3.5 ms to 12 ms. bPYX GFP rats had an increased polysynaptic reflex response (one-way ANOVA F-value = 4.8, p=0.02; bPYX GFP versus naïve, p-value = 0.02). (R) Stimulation of afferents in the radial nerve evoked polysynaptic responses in the (antagonistic) ulnar nerve. (S–T) Representative traces showing recordings from (S) uninjured naïve and (T) bPYX GFP rats. The boxed area highlights polysynaptic compound action potentials which were analysed for *Figure 1U*. (U) The polysynaptic compound action potentials were quantified by measuring the absolute integral (area under the rectified curve) from 3.5 ms to 12 ms. bPYX GFP rats had increased polysynaptic responses versus uninjured naïve rats (one-way ANOVA F-value = 4.2, p=0.03; bPYX GFP versus naïve, p-value = 0.01). (A–U) n = 10 or 11 per group. Data are represented as mean ± SEM.

The following figure supplements are available for figure 1:

**Figure supplement 1.** Lesion cross-sectional areas were similar on the left and right of the medulla.

**Figure supplement 2.** Scoring sheet for spasticity and disordered sensori-motor control of the forelimb.

**Figure supplement 3.** The H reflex undergoes frequency-dependent depression in uninjured naïve rats whereas this is attenuated in rats with bilateral pyramidotomy.

**Figure supplement 4.** Polysynaptic reflex responses recorded from the radial nerve after ulnar nerve stimulation were not changed after injury.

medulla, which confirmed that corticospinal tract fibers running ventrally through the pyramids had been cut (*Figure 1B*; *Figure 2—figure supplement 1*).

Twenty four hours after injury, rats were randomized to treatment with AAV1 encoding either human Neurotrophin-3 or GFP (see below). In the first part of this paper, we describe the forelimb spasticity, other behavioural deficits and changes in spinal reflexes that were observed in rats with bilateral pyramidotomies (and treated with AAV-GFP) compared to uninjured naïve rats. In the second part, we describe how Neurotrophin-3 treatment reduced spasticity, and normalised reflexes, which was accompanied by anatomical changes and behavioural recovery.

Behaviourally, rats with bilateral pyramidotomy developed spasticity in their forelimbs within two weeks of injury (Compare *Video 1* which shows an uninjured naïve rat which exhibited no spasticity with *Video 2* and *3*). The joint movements of the forelimb appeared rigid and the trajectories of the limb were often changed during the swing phase. Rats had narrow stepping with frequently crossed placement of their forepaws. During the loading response of the stance phase we observed vertical oscillatory-like movements, which resulted in bounce-like stepping. Rats displayed spontaneous

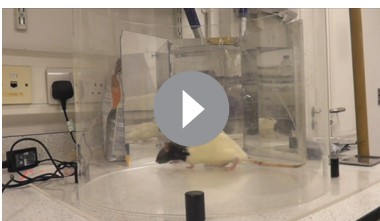

**Video 1.** Open-field movements shown by an uninjured naïve rat (Supplementary to *Figure 1* and Experimental Procedures). Rats are placed in a 50 cm diameter Plexiglas cylinder and videotaped for 3 min every fortnight. See Supplementary *Figure 1—figure supplement 2* for scoring system. During swing, forepaw digits are extended. Joint movements in the forelimb are smooth and linear. Forepaws are placed slightly medial to the shoulders and the loading response in stance phase is normal, i.e. stance initiated with a single placing movement with immediate weight bearing. When rearing without wall contact, the rats regulate their balance fully through the hind-paws and tail; distal forelimbs are held parallel to the floor. During swing, hindpaws are not raised above the lowest point of the belly. Tail twitches are not seen.

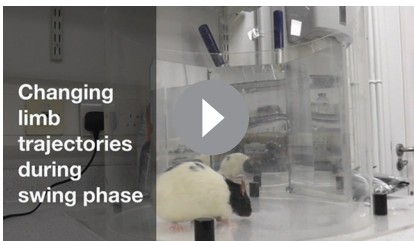

**Video 2.** Open-field scoring of spasticity and disordered sensorimotor forelimb movements of rats with bilateral pyramidotomies (Supplementary to *Figure 1* and Experimental Procedures). Rats exhibiting signs of disordered sensorimotor control have their forepaw digits in a flexed position during swing phase presumably because of hypertonic flexor muscles in the forepaw. Movements of the wrist, elbow and shoulder are corrective (swing trajectory is non-linear) and rigid during swing phase. Stance is narrower compared to controls, with forepaws aligning rostrocaudally or with forepaws crossing over. Furthermore, during the loading response of the stance phase, the forepaw makes multiple contacts as weight bearing starts (muscle jerks causing a 'dampened bounce'). Rats display 'associated reactions', such as single muscle jerks, prolonged muscle contractions and repeated muscle jerks. All these behaviours were scored with the forelimb scale shown in *Figure 1—figure supplement 2*. Videos show a number of different rats at a range of time points after CNS injury.

abnormal prolonged muscle contractions (spasms), and single or repeated muscle jerks (twitch and clonus respectively) with their forelimbs. We observed that spontaneously occurring spasms were more prevalent in forelimb pro-gravity (limb flexor) muscles than anti-gravity (limb extensor) muscles (*Figure 1D*). Their hindlimbs and tail also displayed signs of spasticity (*Video 3*). During the swing phase, rats with bilateral pyramidotomies lifted their hindlimbs higher than uninjured rats, which may indicate increased hip pro-gravity (flexor) muscle activity. Based on our observations, we developed a novel open-field scoring system for rats, which measures spasticity and disordered sensorimotor control of the forelimbs during locomotion and spasticity-associated reactions (*Figure 1—figure supplement 2*; compare *Video 1* with *Video 2*). Using this scoring system we show that spasticity developed by two weeks and was sustained up until the end of the testing period at 10 weeks after bilateral pyramidotomy (*Figure 1E*).

To assess skilled locomotor deficits, we tested rats with bilateral pyramidotomies on the horizontal ladder with irregularly spaced rungs. Rats frequently made errors with their forelimbs because of overstepping or muscle jerks (*Figure 1F*). Furthermore, forelimb grip strength was reduced (*Figure 1G*). Von Frey testing of the forepaw pad revealed very modest mechanical hypersensitivity after injury (*Figure 1H*). Dexterity assessed with the Montoya staircase pellet-reaching test was persistently reduced (*Figure 1I*).

## Rats developed exaggerated low threshold spinal reflexes after bilateral pyramidotomy

To determine the mechanisms whereby bilateral corticospinal tract injury causes spasticity, we assessed spinal reflex excitability with two electrophysiological paradigms using low intensity nerve stimulation while rats were anaesthetized. These revealed hyper-excitability of a forelimb H-reflex and of polysynaptic spinal reflexes in rats with bilateral pyramidotomies.

To measure a proprioceptive Hoffman (H) reflex (*Toda et al., 2014*) we recorded electromyograms from an affected hand muscle (the abductor digiti quinti) whilst stimulating its ulnar nerve (*Figure 1J*). An early M-wave is evoked by motor axon excitation and a later H-wave is evoked by excitation of Ia afferents synapsing onto motor neurons (*Figure 1—figure supplement 3B*). The

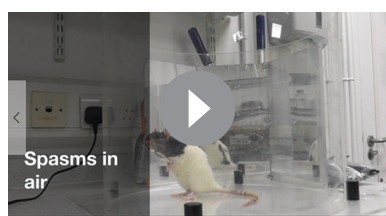

**Video 3.** Additional signs and associated features of spasticity and disordered sensorimotor movements (Supplementary to *Figure 1* and Experimental Procedures). The behaviours shown in this video were not scored as part of the forelimb scale, but were frequently observed. Rats with bilateral pyramidotomy displayed prolonged forelimb muscle contractions and repeated muscle jerks when rearing. Furthermore, stepping with hindlimbs is abnormally high during swing phases (likely to due to increased activation of iliopsoas and other hip flexors) and the hindlimb heels are frequently incompletely placed down. Rats have tail twitches and are unstable during rearing.

H-wave was often evoked at lower stimulation intensities than the M-wave, consistent with work by others (*Tan et al., 2012*). To confirm that the putative H-wave is dependent on sensory afferent stimulation, in three uninjured naïve rat we cut six ipsilateral cervical dorsal roots, which abolished the H-wave and did not affect the M-wave (data not shown). *n.b.*, F-waves are evoked in the rat only at high stimulation intensities and have very low amplitudes (*Meinck, 1976*; *Gozariu et al., 1998*).

The forelimb H-reflex paradigm allowed us to record the H-wave from the same rats prior to injury and then at multiple times after injury. In uninjured naïve rats, when two stimuli were delivered to the ulnar nerve in close succession (for example using an inter-stimulus interval of 0.5 s) then the second H wave (coloured red in *Figure 1—figure supplement 3B*) was smaller than the first H wave (coloured grey in *Figure 1—figure supplement 3B*), which is known as frequency-dependent-depression. In uninjured rats, the second H-wave was depressed at inter-stimulus-intervals of 5 s and less (*Figure 1K*; *Figure 1—figure supplement 3B*). One day after bilateral corticospinal tract injury, the H-wave depression was comparable to before injury (*Figure 1—figure supplement 3D*). However, at two weeks after bilateral pyramidotomy, the H-wave was less depressed at intervals less than 5 s (*Figure 1K*; *Figure 1—figure supplement 3C*). The attenuation of H-wave depression at short inter-stimulus intervals lasted at least up to 10 weeks (*Figure 1—figure supplement 3E*), which suggests there were long lasting changes in synaptic transmission. Frequency-dependent-depression was assessed fortnightly. We measured the area under the stimulus-response curve from inter-stimulus intervals of 10 s to 0.1 s (*Figure 1—figure supplement 3F*) for each rat at each time point separately and then plotted the group mean values (*Figure 1L*). Uninjured naïve rats showed consistent frequency-dependent depression of the H-wave over 10 weeks, but injured control rats had reduced frequency-dependent depression from 2 weeks up until the end of the testing period at 10 weeks (*Figure 1L*, *Table 1*), which is evidence for changed synaptic transmission from Ia afferent fibers to motor neurons.

We next determined whether polysynaptic spinal reflexes that can produce compensatory responses during movement (*Dietz, 2002*) are also affected by injury. Ten weeks after bilateral corticospinal tract injury, rats were terminally anesthetized and the ulnar, median and radial nerves were exposed on the treated side (*Figure 1M*) as described previously (*Bosch et al., 2012*). We studied three different reflexes:

Firstly, we recorded from the ulnar nerve whilst stimulating the median nerve (*Figure 1N*). Both carry afferents from and innervate synergist forelimb flexor muscles which received AAV injections (magenta muscles in *Figure 1M*). Low intensity and low frequency stimulation activates large diameter neurons that include group Ia, Ib and II muscle afferents and group I range skin afferents but not high threshold ones such as nociceptive C fibers (*Brock et al., 1951*; *Nakatsuka et al., 2000*). Low intensity, low frequency stimulation of the median nerve evoked polysynaptic compound action potentials in the ulnar nerve in uninjured naïve rats (*Figure 1O*). They were increased 10 weeks after bilateral corticospinal tract injury indicating a hyper-excitability of polysynaptic spinal reflexes (*Figure 1P,Q*).

Next, we recorded from the ulnar nerve whilst stimulating its antagonistic radial nerve (*Figure 1R*). Low intensity and low frequency radial nerve stimulation evoked only very few to no polysynaptic compound action potentials in the antagonistic ulnar nerve in uninjured naïve rats (*Figure 1S*). Ten weeks after bilateral pyramidotomy, polysynaptic compound action potentials were increased (*Figure 1T,U*) indicating loss of specificity in afferent connectivity (Figure 6). This gain of aberrant connectivity may cause increased abnormal co-contraction of antagonists. In summary, after

**Table 1.** Properties of the M-wave and H-reflex (Supplementary to **Figures 1** and **3**). Values (mean ± SEM) are given for motor threshold, M-wave and H-wave latencies and maximum amplitudes, maximum depression of H-wave. The maximum depression of the H-wave is different between bPYX GFP and bPYX NT3 animals (n = 10 to 11/group, Two-way RM ANOVA, group F = 6.9, p<0.001, group*week F = 2.3 p = 0.047; post-hoc analysis revealed differences between uninjured naïve rats and bPYX GFP or bPYX NT3 at Week 2, bPYX GFP and bPYX NT3 or uninjured naïve rats at Week 6, 8 and 10, Fisher's LSD, p-values *<0.05, **<0.01, ***<0.001, ****<0.0001).

| | | baseline | Week 2 | Week 4 | Week 6 | Week 8 | Week 10 |
|---|---|---|---|---|---|---|---|
| Motor threshold (mA) | naïve | 1.14 ± 0.2 | 2.25 ± 0.5 | 2.13 ± 0.5 | 2.01 ± 1.5 | 2.27 ± 0.4 | 3.45 ± 0.5 |
| | bPYX GFP | 1.83 ± 0.3 | 1.26 ± 0.2 | 1.71 ± 1.4 | 2.42 ± 0.5 | 2.15 ± 0.4 | 2.05 ± 1.6 |
| | bPYX NT3 | 1.28 ± 0.2 | 1.33 ± 0.3 | 2.06 ± 0.4 | 1.62 ± 0.3 | 1.79 ± 0.5 | 2.04 ± 0.5 |
| M-wave latency (ms) | naïve | 0.96 ± 0.0 | 0.95 ± 0.1 | 0.85 ± 0.0 | 0.88 ± 0.0 | 0.97 ± 0.1 | 1.13 ± 0.1 |
| | bPYX GFP | 0.92 ± 0.1 | 0.91 ± 0.1 | 0.98 ± 0.0 | 0.99 ± 0.1 | 0.91 ± 0.0 | 1.01 ± 0.1 |
| | bPYX NT3 | 0.99 ± 0.1 | 0.89 ± 0.0 | 1.24 ± 0.3 | 0.98 ± 0.1 | 0.94 ± 0.1 | 1.12 ± 0.1 |
| maximum M-wave (mV) | naïve | 7.23 ± 0.9 | 6.76 ± 0.5 | 5.53 ± 0.4 | 4.76 ± 0.5 | 4.83 ± 0.5 | 3.31 ± 0.4 |
| | bPYX GFP | 6.34 ± 0.8 | 6.94 ± 0.9 | 8.34 ± 0.7 | 5.84 ± 0.7 | 6.92 ± 0.6 | 4.93 ± 0.7 |
| | bPYX NT3 | 7.89 ± 0.8 | 6.25 ± 0.7 | 4.82 ± 1.0 | 6.08 ± 0.7 | 3.74 ± 0.4 | 4.32 ± 0.5 |
| H-wave latency (ms) | naïve | 5.66 ± 0.1 | 5.60 ± 0.1 | 5.39 ± 0.1 | 5.42 ± 0.1 | 5.69 ± 0.1 | 5.97 ± 0.1 |
| | bPYX GFP | 5.48 ± 0.1 | 5.24 ± 0.1 | 5.67 ± 0.1 | 5.55 ± 0.4 | 5.42 ± 0.1 | 5.60 ± 0.2 |
| | bPYX NT3 | 5.22 ± 0.1 | 5.30 ± 0.2 | 5.80 ± 0.3 | 5.48 ± 0.1 | 5.36 ± 0.2 | 5.40 ± 0.1 |
| maximum H-wave (mA) | naïve | 1.34 ± 0.2 | 1.46 ± 0.3 | 1.09 ± 0.2 | 1.22 ± 0.2 | 1.31 ± 0.2 | 0.92 ± 0.2 |
| | bPYX GFP | 1.60 ± 0.3 | 3.50 ± 0.7 | 2.84 ± 0.5 | 2.48 ± 1.8 | 2.72 ± 0.6 | 1.94 ± 0.5 |
| | bPYX NT3 | 2.43 ± 0.5 | 2.15 ± 0.4 | 1.53 ± 0.3 | 2.01 ± 0.3 | 1.41 ± 0.2 | 1.63 ± 0.2 |
| maximum H-wave (%) | naïve | 19.5 ± 8.0 | 21.7 ± 3.3 | 19.7 ± 3.5 | 29.9 ± 6.5 | 26.5 ± 3.0 | 31.2 ± 5.8 |
| | bPYX GFP | 29.5 ± 6.7 | 46.2 ± 4.7 | 31.9 ± 4.8 | 41.8 ± 5.7 | 40.8 ± 6.2 | 43.8 ± 5.5 |
| | bPYX NT3 | 30.7 ± 5.7 | 38.2 ± 6.5 | 36.8 ± 4.8 | 44 ± 10.7 | 45.3 ± 9.1 | 45.0 ± 8.8 |
| maximum depression of H-wave | naïve | 24.3 ± 4.2 | 35.9 ± 3.8 **** | 42.7 ± 6.7 | 37.9 ± 7.3 | 34.7 ± 6.6 | 41.0 ± 5.5 |
| | bPYX GFP | 38.5 ± 5.9 | 56.4 ± 7.9 | 57.5 ± 8.7 | 61.0 ± 8.c2 *** | 56.3 ± 8.1 * | 62.9 ± 8.3 *** |
| | bPYX NT3 | 37.9 ± 5.9 | 58.9 ± 6.1 | 48.6 ± 8.0 | 37.6 ± 5.8 | 36.5 ± 6.3 | 29.2 ± 3.9 |

Table 1: Properties of the M- and H-wave.

bilateral pyramidotomy, stimulation of agonist (median) nerves revealed exaggerated spinal reflexes in the ulnar nerve (**Figure 1M–Q**) whilst stimulation of the homonymous (ulnar) nerve revealed exaggerated spinal reflexes to a hand flexor muscle (**Figure 1J–L**). These findings demonstrate spinal motor circuits of pro-gravity limb flexor muscle groups were hyper-excitable after injury, which is supported by our behavioural observation that flexor muscles were more likely to be affected by spasms compared to extensors (**Figure 1D**).

Lastly, we recorded from the radial nerve while stimulating the ulnar nerve (**Figure 1—figure supplement 4A**). Low intensity and low frequency ulnar nerve stimulation rarely evoked polysynaptic compound action potentials in the antagonistic radial nerve in uninjured naïve rats (**Figure 1—figure supplement 4B**) and in rats with bilateral pyramidotomy (**Figure 1—figure supplement 4C,D**). Thus, we did not observe any or increased reflex activity in the extensor radial nerve after antagonist (ulnar) stimulation after injury (**Figure 1—figure supplement 4**). Consistent with this, we seldom observed spasms in forelimb extensor muscles in freely moving rats (**Figure 1D**; **Video 2**). Taken together, bilateral pyramidotomy led to increased reflexes and muscle spasms primarily involving flexor muscles.

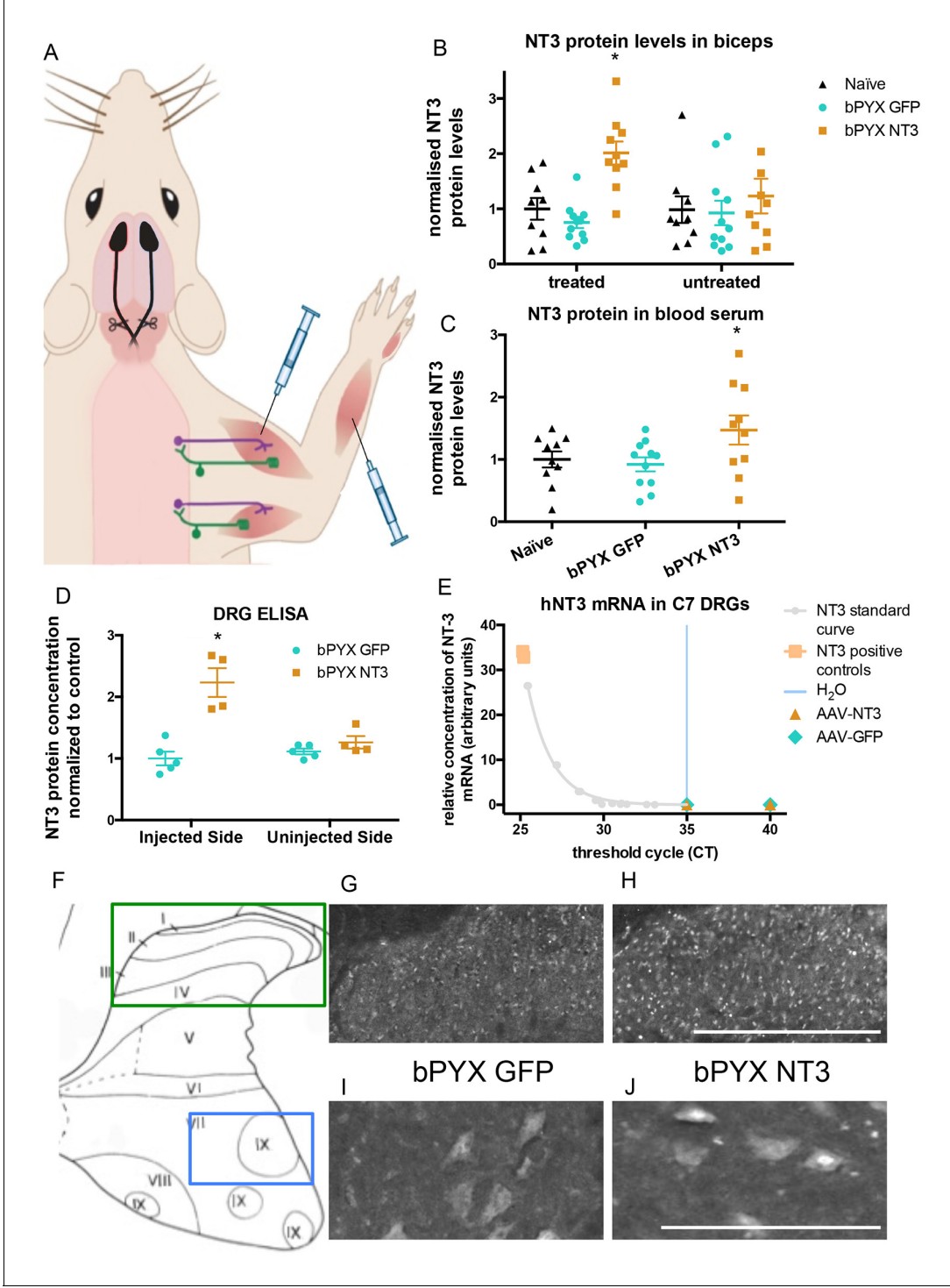

**Figure 2.** Increased levels of neurotrophin-3 in muscle and blood post-injection of AAV1 neurotrophin-3 into forelimb muscles. (**A**) Schematic showing experimental set-up. Rats received bilateral pyramidotomies and were injected either AAV1-NT3 or AAV1-GFP into forelimb muscles. The monosynaptic proprioceptive reflexes are shown. Group Ia afferents project from the muscle via their cell bodies in the DRG into the spinal cord. Motor neurons project to the muscles. (**B**) ELISAs confirmed that neurotrophin-3 is upregulated in the *biceps brachii* 10 weeks post-injection with AAV1-NT3 (n = 10/11 per group, RM two-way ANOVA, group F = 7.2, p = 0.003, post-hoc analysis revealed an increase only on the injected side of the bPYX NT3 group versus naïve or bPYX GFP, p-values<0.05). (**C**) The Neurotrophin-3 level in the blood serum was also increased after AAV1 NT-3 treatment (one-way ANOVA, F-value = 3.4, p=0.047, bPYX NT3 versus naïve or bPYX GFP, p-values<0.05). (**D**) A separate cohort of rats was injected with either AAV1-GFP or AAV1-NT3 into forelimb muscles. ELISAs of C3 to C8 DRGs showed a 2.3 fold increase of NT3 only in DRGs from the side ipsilateral 4 weeks after injections (n = 4/5 per group, one-way ANOVA, F = 34.13, p<0.001, AAV-NT3 injected side versus all other groups, p-values<0.05). (**E**) qRT-PCR
*Figure 2 continued on next page*

*Figure 2 continued*

for human NT-3 mRNA confirms that the viral vector is not transported retrogradely to the DRG. Human neurotrophin-3 mRNA was detected in positive control samples (human brain cDNA) and in the standard curve but not detected in the ipsilateral (left) DRGs at 4 weeks after injection of AAV-NT3 or AAV-GFP (n = 5 per group). The x-axis shows the cycle at which signal rose above background threshold ($C_T$) versus the y-axis which shows the concentration of human NT-3 cDNA with a standard curve plotted through points of the standard. Samples from AAV-NT3 and AAV-GFP groups had $C_T$ values $\geq 35$ (comparable to water as No Template Control), indicating absence of human NT-3 mRNA in those samples. (**F–J**) Transverse cervical spinal cord sections were immunostained for NT3. There was increased immunoreactivity in dorsal horn neurons (compare **H** with **G**) and in the nuclei of motor neurons in laminae XI (compare **J** with **I**) in the AAV-NT3 group versus AAV-GFP group although we did not detect increased levels of NT3 protein in the spinal cord overall, see *Figure 2—figure supplement 2*. Scale bars 0.5 mm. (**B–E**) Data are represented as mean ± SEM.

The following figure supplements are available for figure 2:

**Figure supplement 1.** Lesion cross-sectional areas were similar on the left and right of the medulla in both bPYX groups.

**Figure supplement 2.** No increased levels of neurotrophin-3 in homogenates of *triceps brachii*, spinal cord or liver after injections of an AAV expressing neurotrophin-3 into the *biceps brachii* and other forelimb flexors.

## Intramuscular injection of AAV1-Neurotrophin-3 increased levels of Neurotrophin-3 protein in treated flexor muscles, ipsilateral cervical DRG and ipsilateral spinal cord neurons

Twenty four hours after bilateral pyramidotomy, rats were randomized to treatment with AAV1 encoding either human Neurotrophin-3 or GFP, which was injected into many forelimb flexor muscles (listed in Methods) (*Figure 2A*). At the end of the study, Enzyme-Linked Immunosorbent Assays (ELISAs) showed that Neurotrophin-3 protein levels were elevated in treated muscles (e.g., in the ipsilateral but not contralateral biceps brachii) (*Figure 2B*) as well as in the blood (*Figure 2C*). There was a non-significant trend toward increased Neurotrophin-3 in the ipsilateral triceps brachii (*Figure 2—figure supplement 2*) which may have resulted from AAV spread from nearby muscles (e.g., biceps brachii) during injection or transport in the bloodstream. We did not detect any trend to an increase contralaterally in muscles (*Figure 2*, *Figure 2—figure supplement 2*).

Homogenates of cervical DRGs were used for RNAseq (data not shown) and the spinal cords were processed for immunohistochemistry (see below). We generated an additional cohort of rats to measure NT-3 protein levels in cervical DRGs, the spinal cord and other tissues at 4 weeks after injection of AAV1. ELISA showed that Neurotrophin-3 protein levels were elevated in ipsilateral (but not contralateral) cervical C3-C8 DRGs (*Figure 2D*), consistent with retrograde transport of hypothetically either AAV1 particles and/or Neurotrophin-3 protein itself. We found that the AAV1 vector carrying the transgene was not retrogradely transported to the DRG (*Figure 2E*). RNA from homogenates of cervical DRGs were subjected to qRTPCR using primers that distinguish human Neurotrophin-3 mRNA (i.e., transgene specific) from rat Neurotrophin-3 mRNA (i.e., endogenous). Samples from AAV-NT3 and AAV-GFP groups had $C_T$ values $\geq 35$, comparable to the No Template Control (water). In other words, we did not detect human Neurotrophin-3 mRNA in the ipsilateral or contralateral cervical DRGs of rats injected with AAV1-NT3 or AAV1-GFP (*Figure 2E*) although we did detect a strong signal for human Neurotrophin-3 in positive control samples of human brain cDNA. Together these results show that the viral vector itself was not retrogradely transported to the DRG, but the neurotrophin-3 protein itself was.

Intramuscular treatment using AAV1-NT3 in this cohort did not significantly increase levels of Neurotrophin-3 protein in homogenates of ipsilateral or contralateral spinal cervical hemicords (*Figure 2—figure supplement 2B*) or liver (*Figure 2—figure supplement 2C*) 4 weeks after injection. We next used a more sensitive method, namely immunolabeling, to see whether Neurotrophin-3 was detectable in neurons of the dorsal or ventral horns of the cervical spinal cord (*Figure 2F–J*). Immunolabeling revealed that, in rats treated with AAV1-NT3, Neurotrophin-3 protein was increased in neurons of the ipsilateral cervical dorsal horn (*Figure 2H*) and ipsilateral cervical motor neurons (including in their nuclei) (*Figure 2J*) relative to AAV1-GFP (*Figure 2G,I* respectively) but not in contralateral dorsal horn or motor neurons. Together these data are consistent with other studies showing receptor-mediated retrograde transport of Neurotrophin-3 protein from injected muscles in large diameter sensory neurons and motor neurons (*Distefano et al., 1992*; *Helgren et al., 1997*).

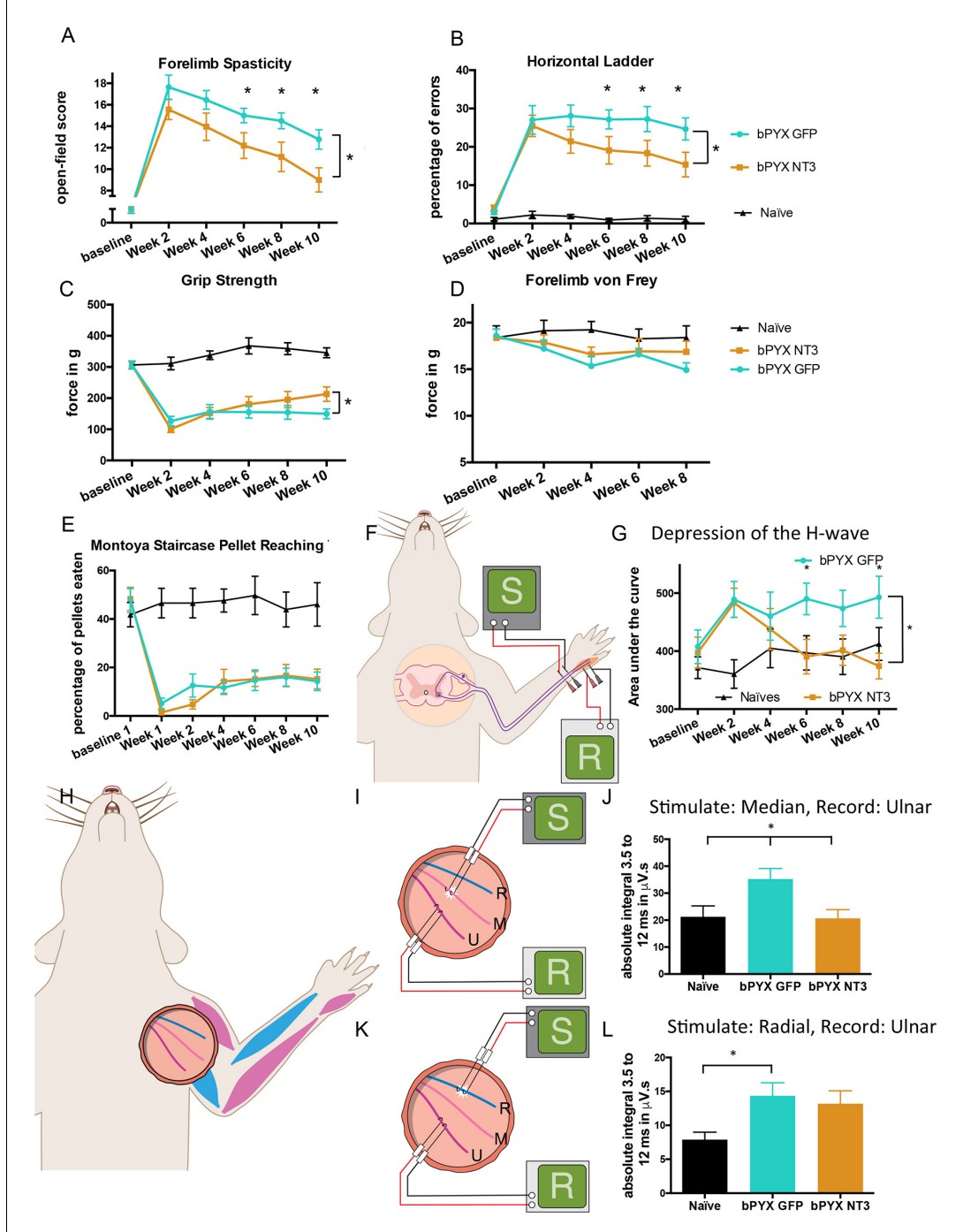

**Figure 3.** Intramuscular Neurotrophin-3 treatment improved functional recovery and reduced spasms after bilateral pyramidotomy. Please note that for clarity in describing our model of spasticity, *Figure 1* contained information from *Figure 3* relating to the uninjured naïve and bPYX GFP groups. (A) Neurotrophin-3 treatment reduced signs of spasticity (RM two-way ANOVA, group F = 19.8, p<0.001; bPYX NT3 versus bPYX GFP at Week 6, 8 and 10, p-values<0.05). (B) Neurotrophin-3 caused rats to make fewer errors on the horizontal ladder with their treated forelimb as a percentage of the total steps taken (RM two-way ANOVA, group F = 123.4, p<0.001; bPYX NT3 versus bPYX GFP p<0.0001; bPYX NT3 versus bPYX GFP at Week 6, 8 and 10, p-values<0.05). (C) Unilateral Grip Strength Test. Neurotrophin-3 treatment slightly improved grip strength of the treated forepaw at 10 weeks (RM two-way ANOVA, group F = 145.0, p<0.001; bPYX NT3 vs bPYX GFP p = 0.11; bPYX NT3 vs bPYX GFP at Week 10, p-value = 0.026). (D) Cutaneous mechanical hypersensitivity was not affected by NT3 treatment assessed using the automated von Frey test (RM two-way ANOVA, group F = 5.2 p=0.019; bPYX NT3 versus bPYX GFP p = 0.29). (E) Dexterity was assessed using the staircase test. The two treatment groups were impaired relative to uninjured naïve rats, and no differences were detected between two treatment groups post-injury (RM two-way ANOVA, group F = 100.3, p<0.0001; bPYX NT3 versus bPYX GFP p = 0.69) confirming that the corticospinal tract is essential for recovery of fine motor function (*Weidner et al., 2001*). (F)

*Figure 3 continued on next page*

*Figure 3 continued*

Schematic showing the H-reflex paradigm. The ulnar nerve was stimulated distally and EMGs were recorded from a homonymous hand muscle (abductor digiti quinti). (G) Graph shows changes over time in hyper-reflexia, measured as the area under the curve of the frequency-dependent depression (*Figure 1—figure supplement 3F*). Injury groups had exaggerated reflexes from 2 weeks post-injury, but NT3 treated rats had normal H-wave depression from 6 weeks onwards. (RM two-way ANOVA, group F = 5.9 p<0.001; bPYX NT3 versus bPYX GFP p = 0.024; bPYX NT3 versus bPYX GFP at Week 6 and 10, p-values<0.05). (H) At week 10, the radial, median and ulnar nerves were exposed for stimulation and recording. The radial nerve (blue) innervates extensor muscles (blue) whereas the median and ulnar nerves (pink and magenta) innervate synergist flexor muscles that were injected with AAV (pink and magenta). (I) Stimulation of afferents in the median nerve evoked responses in the (synergist, flexor) ulnar nerve (J) The polysynaptic compound action potentials were quantified by measuring the absolute integral (area under the rectified curve) from 3.5 ms to 12 ms. Neurotrophin-3 treatment restored the exaggerated reflexes to normal (one-way ANOVA F-value = 4.8, p=0.02, bPYX NT3 versus bPYX GFP, p-values = 0.01) (K) Stimulation of afferents in the radial nerve evoked polysynaptic responses in the (antagonistic) ulnar nerve. (L) The polysynaptic compound action potentials were quantified by measuring the absolute integral (area under the rectified curve) from 3.5 ms to 12 ms. Neurotrophin-3 treatment did not restore the exaggerated reflexes to normal (one-way ANOVA F-value = 4.2, p=0.03, bPYX NT3 versus bPYX GFP, p-values = 0.63) (A–L) n = 10 or 11 per group. Data are represented as mean ± SEM.

The following figure supplement is available for figure 3:

**Figure supplement 1.** Changes in polysynaptic reflexes after bilateral pyramidotomy and with Neurotrophin-3.

In summary, Neurotrophin-3 protein levels were increased in ipsilateral flexor muscles, ipsilateral cervical DRG and in the ipsilateral spinal cord as well as in blood.

## Treatment with Neurotrophin-3 improved functional outcomes after bilateral pyramidotomy

We assessed the forelimb behaviour of rats treated with either AAV-NT3 or AAV-GFP with our newly developed open-field scoring system (*Figure 3A*). Within two weeks after bilateral pyramidotomy injury both treatment groups developed spasticity in their forelimbs. Neurotrophin-3 treatment led to a reduction of forelimb spasticity which was still ongoing at 10 weeks post-injury and treatment (*Figure 3A*).

To assess locomotor deficits, we tested rats on the horizontal ladder task. Rats with bilateral pyramidotomy treated with AAV-GFP frequently missed rungs, but intramuscular Neurotrophin-3 treatment gradually improved precision stepping on the horizontal ladder throughout the 10 weeks testing period (*Figure 3B*). Additionally, Neurotrophin-3 treated rats partially recovered grip strength with their treated forelimb (*Figure 3C*). Importantly, Neurotrophin-3 did not cause or affect mechanical hypersensitivity, as expected from human clinical trials (*Parkman et al., 2003*; *Sahenk, 2007*; *Chaudhry et al., 2000*; *Coulie et al., 2000*; *Sahenk et al., 2005*) (*Figure 3D*). Intramuscular AAV-NT3 treatment did not improve recovery on the pellet reaching task after bilateral pyramidotomy (*Figure 3E*) which is consistent with the fact that corticospinal injury leads to a persistent deficit in dexterity (*Weidner et al., 2001*). Taken together, these behavioural results demonstrate that bilateral pyramidotomy is sufficient to cause spasticity in rodents and that these symptoms can be improved by treating flexor muscles with Neurotrophin-3. Treatment also resulted in improved functional recovery of motor behaviour. We next show that NT3 modifies neurophysiological properties of spinal reflexes whose low threshold proprioceptive and cutaneous mechanoreceptive afferents and motor neurons are known to express TrkC (*Lee et al., 2012*; *McMahon et al., 1994*).

## Neurotrophin-3 normalized spinal reflexes evoked by low intensity stimulation of nerves containing afferents from injected muscles

As described above, rats showed less H-wave depression at inter-stimulus intervals of 5 s and less after bilateral pyramidotomy. Intramuscular overexpression of Neurotrophin-3 restored the frequency-dependent depression of the H-wave by 6 weeks post-injury (*Figure 3F,G*, *Table 1*), consistent with injection of AAV-NT3 into this hand muscle.

Neurotrophin-3 treatment also modified polysynaptic spinal reflexes involving afferents from treated muscles. Neurotrophin-3 normalised the exaggerated polysynaptic responses that were measured in the ulnar nerve after median nerve stimulation (*Figure 3I–J*; *Figure 3—figure*

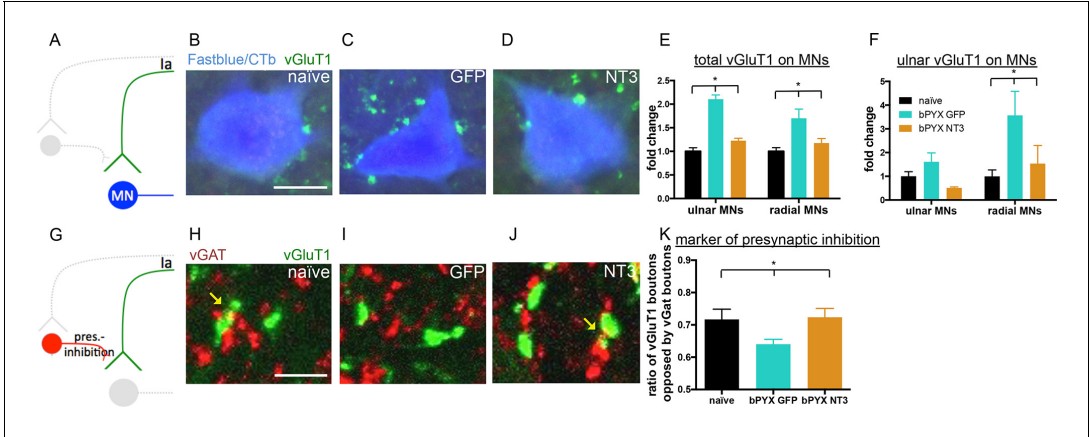

**Figure 4.** Neurotrophin-3 treatment restored balance between excitatory and inhibitory causes of spasticity. (A–D) Ia boutons were identified by vGluT1 immunolabelling (green) and motor neurons were traced retrogradely with Fast Blue or Cholera Toxin beta (blue) in (B) uninjured naïve rats, (C) bPYX GFP rats and (D) bPYX NT3 rats. C8, scale bar: 20 µm. (E) bPYX caused an increase in vGluT1 boutons in close proximity to motor neurons. Neurotrophin-3 reduced the number of boutons to normal (two-way-ANOVA, group F = 33.5 p<0.001; bPYX GFP versus naïve or bPYX NT3, p-values<0.05). (F) The number of vGluT1 boutons of afferents from the ulnar nerve (CTb traced) increased on radial but not ulnar motor neurons after bilateral pyramidotomy, which was normalized with NT3 treatment (RM two-way-ANOVA, group F = 4.6, p = 0.01, motor neurons F = 4.2, p = 0.047; radial MNs bPYX GFP versus naïve or bPYX NT3, p-values<0.05). (G) Proprioceptive afferent boutons (vGluT1, green) receive pre-synaptic inhibition from boutons immunopositive for the vesicular GABA transporter (vGAT, red) in (H) uninjured naïve rats, (I) bPYX GFP rats and (J) bPYX NT3 rats. C8 transverse spinal cord sections, scale bar 10 µm. Yellow arrows indicate putative pre-synaptically inhibited vGluT1 synapses. (K) Injury led to a reduction in vGluT1 boutons receiving presynaptic inhibition whereas NT3 restored levels of presynaptic inhibition (one-way ANOVA, F = 3.6 p = 0.042; bPYX GFP versus naïve or bPYX NT3, p-values<0.05). (E,F,K) n = 10 or 11 per group. Data are represented as mean ± SEM.

The following figure supplement is available for figure 4:

**Figure supplement 1.** The density of inhibitory boutons directly onto motor neurons did not change with injury or neurotrophin-3 treatment.

supplement 1A). Both nerves carry afferents from and innervate synergist forelimb muscles that were treated with an AAV1 overexpressing Neurotrophin-3. However, Neurotrophin-3 treated animals had no reduction of the exaggerated polysynaptic responses recorded in the ulnar nerve after radial nerve stimulation (*Figure 3K–L*; *Figure 3—figure supplement 1B*), consistent with the fact that only flexor muscles were injected with AAV-NT3 and that proprioceptive afferent fibers in the radial nerve (supplying extensor muscles) did not receive direct treatment. Finally, Neurotrophin-3 treatment did not modify polysynaptic responses evoked in the radial nerve after stimulation of the ulnar nerve; these responses did not change after injury either (*Figure 3—figure supplement 1C–D*). Taken together, we conclude that intramuscular Neurotrophin-3 treatment normalized low-threshold spinal reflexes whose afferents were exposed to Neurotrophin-3.

## Neurotrophin-3 treatment normalized the inhibitory and excitatory balance in the spinal cord

We characterised functional aspects of sensorimotor abnormalities and hyperreflexia of the forelimb after CNS injury as well as how Neurotrophin-3 normalizes these. However, the effects of injury and treatment on the anatomical and molecular makeup within the spinal cord remained to be understood. By studying markers of excitatory and inhibitory networks within the spinal cord, we found that Neurotrophin-3 restored these networks to naïve levels.

First, to identify motor neuron groups supplying forelimb flexor and extensor muscles, the ulnar nerve and radial nerve on the treated side were traced retrogradely with either Cholera Toxin beta subunit (CTb) or Fast blue, respectively. Transverse sections of the cervical spinal cord were immunolabeled for vGluT1 which is located in excitatory boutons of proprioceptive and cutaneous mechanosensory axons (*Alvarez et al., 2011*). In the intact rat, vGluT1 boutons on motor neurons are from proprioceptive afferents (*Stepien et al., 2010*; *Betley et al., 2009*). We determined whether the number of vGluT1 immunoreactive boutons in close proximity to motor neurons changed after

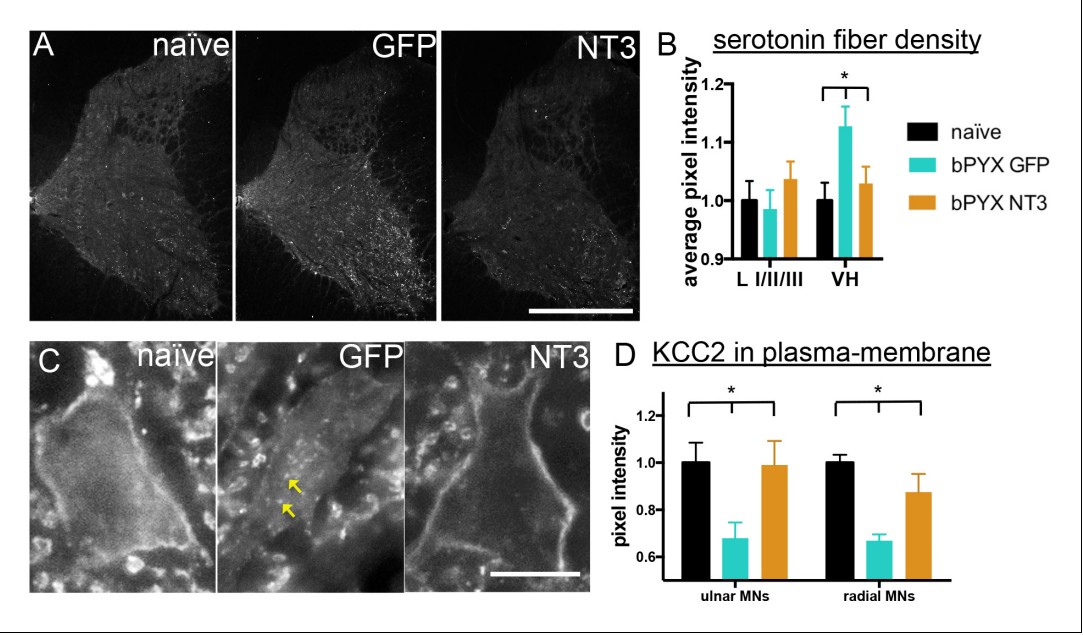

**Figure 5.** Neurotrophin-3 treatment normalised serotonergic innervation of the C7/8 spinal cord and the ion transporter KCC2 in motor neuron membranes to normal. (**A**) Representative images of C7/8 spinal cords on the treated side immunolabelled for serotonin in uninjured naïve, bPYX GFP and bPYX NT3 rats. Scale bar: 1 mm. (**B**) Serotonergic pixel intensity in Ventral Horn (laminae VII-IX) was increased after injury, but NT3 treatment restored this to normal levels (RM two-way ANOVA, group*horn interaction F = 3.5 p = 0.037; Ventral Horn bPYX GFP versus naïve or bPYX NT3, p<0.05). Pixel intensities in the dorsal horn (laminae I-III) did not change. (**C–D**) KCC2 immunolabeling: methods can be found in *Figure 5—figure supplement 1* The pixel intensity of KCC2 across the plasma membrane of motor neurons was decreased after injury but accumulations of KCC2 were seen intracellularly (yellow arrows). KCC2 was normalized by NT3 (RM two-way ANOVA, group F = 10.2, p<0.001; bPYX GFP versus naïve or bPYX NT3, p-values<0.05). Scale bar, 20 μm. (**B, D**) n = 10 or 11 per group. Data are represented as mean ± SEM.

The following figure supplement is available for figure 5:

**Figure supplement 1.** Analysis of KCC2 in the membrane of motor neurons.

bilateral pyramidotomy (*Figure 4A–C*). After injury, more vGluT1 boutons were found in close proximity to cervical motor neurons of the ulnar and radial nerves (*Figure 4E*) which might contribute to the spasticity we report. To explore the specificity of afferent fibers synapsing onto different motor neuron pools, we took advantage of the fact that injection of CTb into the ulnar nerve also transganglionically labels predominantly large diameter afferents (*Shehab and Hughes, 2011*). CTb-immunoreactive afferents from the ulnar nerve had more vGluT1 boutons in close proximity to radial motor neurons (*Figure 4F*). This suggests aberrant strengthening of excitatory afferent connectivity upon antagonists which might mediate increased co-contractions.

Synapses from primary afferents onto motor neurons (i.e., proprioceptive afferents) are often modulated by inhibitory synapses from GABAergic interneurons onto the afferent axon terminals (*Figure 4G*) (*Fink et al., 2014*; *Betley et al., 2009*). We immunolabeled sections of the cervical spinal cord for vGAT (to label presynaptic inhibitory boutons) and for vGluT1 (*Figure 4H–I*). Analysis of z-stack confocal images revealed vGAT immunoreactive boutons in close proximity to vGluT1 immunoreactive boutons in the ventral horn. After bilateral pyramidotomy, there were fewer vGAT boutons in close proximity to vGluT1 positive boutons (*Figure 4K*) suggesting they received less presynaptic inhibition by GABAergic interneurons. Together, these changes indicate abnormally increased excitation of motor neurons by agonist and antagonist proprioceptive afferents and explain the hyper-reflexia, co-contraction and spasms exhibited by rats with bilateral corticospinal tract injuries.

Intramuscular Neurotrophin-3 normalized the number of excitatory vGluT1 boutons onto ulnar and radial motor neurons (*Figure 4D–E*), reduced CTb+ ulnar excitatory boutons near to radial motor neurons towards the levels seen in uninjured naïve rats (*Figure 4F*) and increased apparent

presynaptic inhibition of vGluT1 boutons to normal levels (*Figure 4J–K*). We propose these changes contribute to the normalization of the spinal reflexes described above. The number of vGAT immunoreactive inhibitory boutons upon motor neurons which mediate post-synaptic inhibition was not affected by injury or Neurotrophin-3 treatment (*Figure 4—figure supplement 1*). Together, these results indicate that Neurotrophin-3 treatment normalised the balance of excitatory and inhibitory inputs to motor neurons.

Changes in the serotonergic system can also cause spasticity after spinal cord injury. Complete transection causes an up-regulation of the constitutively active $5HT_{2C}$ receptor (*Murray et al., 2010*) and incomplete spinal cord injury leaves spared serotonergic fibers in the sensitized spinal cord (*D'Amico et al., 2013*) which both can cause spasticity by enhancing persistent inward currents. Cervical spinal cord sections were immunolabeled for serotonin and we measured the pixel intensity in the dorsal laminae I/II/III and the ventral horn. Corticospinal tract injury resulted in an increase of serotonergic fiber density in the ventral horn, but not in the dorsal horn; the former was normalized by Neurotrophin-3 treatment (*Figure 5A,B*). In the ventral horn, serotonin receptor activity can cause hyper-excitability of motor neurons by regulating the ion symporter KCC2 (*Bos et al., 2013*). KCC2, when located to the membrane, expels chloride$^-$ and potassium$^+$ ions to regulate the reversal potential of $GABA_A$ responses; if membrane KCC2 is reduced, motor neurons become hyper-excitable (*Boulenguez et al., 2010*). After injury, we found less KCC2 in the membrane of ulnar and radial motor neurons and more accumulations within the somata (*Figure 5C,D*). Neurotrophin-3 treatment translocated KCC2 back to the membrane of both ulnar and radial motor neurons (*Figure 5C,D*).

In conclusion, we demonstrated that afferent fibre patterning and motor neuron properties, which both project to muscles, were altered with intramuscular Neurotrophin-3 treatment after bilateral corticospinal tract injury. We have shown that Neurotrophin-3 normalises mono- and poly-synaptic low threshold reflexes and leads to improved motor recovery after pyramidotomy lesions.

## Discussion

To examine underlying causes of spasticity, we developed a novel rodent model of spasticity, which displays several features of the human condition. We found that excitatory networks within the spinal cord were upregulated after bilateral pyramidotomy and proprioceptive afferents had increased targeting of antagonistic motor neurons more than in normal rats. We showed that bilateral pyramidotomy caused hyperreflexia, abnormal sensorimotor processing, spasms and clumsiness during walking. Interestingly, behavioural, neurophysiological and anatomical abnormalities could be dramatically improved with delayed intramuscular Neurotrophin-3 treatment, and we showed that Neurotrophin-3 overexpression in the muscle restored accurate patterning of central connectivity and the amount of KCC2 in the motor neuron membrane which is important for normal excitability.

### Bilateral corticospinal tract lesioning is sufficient to cause spasticity

Our rodent model gives evidence that selective and complete lesioning of the corticospinal tract results in spasticity. From the clinicoanatomic perspective, there is an extensive debate whether selective pyramidal tract damage (a rare event) is sufficient to cause spastic hemiplegia in humans. Stroke and spinal cord injury, which may or may not affect the corticospinal tract, very commonly lead to spasticity (*Adams and Hicks, 2005*). Spasticity symptoms have sometimes been ascribed to additional damage to extrapyramidal structures, but a recent extensive review of pathological cases gives evidence that damage to the pyramidal tracts is sufficient and necessary to cause spastic hemiplegia in humans (*de Oliveira-Souza, 2015*). The classic lesion experiments by Lawrence and Kuypers revealed the role of the corticospinal tract in motor control in non-human primates (*Lawrence and Kuypers, 1968a*, *1968b*). A large cohort of macaque monkeys received bilateral corticospinal tract injuries in the pyramids. While skilled hand function is severely affected by the lesions, the monkeys were still able to walk and climb shortly after the injury. We re-examined their original videos (*Lemon et al., 2012*) and observed clonic movements (their Supplementary Videos 2, 3 and 8), narrowed stance during locomotion (their Supplementary Video 6) and increased flexor activities in their forelimbs (their Supplementary Video 7) supporting the notion that corticospinal tract injury is sufficient to produce spastic signs and abnormal locomotion in non-human primates as well as humans.

Other groups have modelled spasticity in rodents, which has advanced the understanding of underlying causes of spasticity and the development of novel potential therapies (*Gonzenbach et al., 2010*; *Boulenguez et al., 2010*; *Corleto et al., 2015*). After thoracic cord transection or contusion, rats show lower limb spasticity (*Côté et al., 2011*; *Thompson et al., 1998*; *Bose et al., 2002*; *Corleto et al., 2015*; *Yates et al., 2008*) and after sacral cord transection rodents displayed tail spasms (*Bennett et al., 2004*; *Kapitza et al., 2012*; *Murray et al., 2010*). A midthoracic T-lesion, which essentially damages the dorsomedial, dorsolateral and ventromedial parts of the corticospinal tracts, results in hindlimb spasms during swimming (*Gonzenbach et al., 2010*). Whereas these injuries provide models of hindlimb/tail spasticity, which can be measured after stimulation or during specialized behavioural testing for rats, our model of bilateral pyramidotomy allows observation of spasticity in the forelimbs, hindlimbs and tail during awake, free movements in the open field. Our model shows visible signs of spasticity including increased reflexes similar to the human condition. Our model provides further evidence that exclusive corticospinal tract lesioning is sufficient to produce signs of spasticity. Our model also has allowed us to identify a new intervention to treat the underlying neuronal component of disordered sensorimotor control resulting from upper motor neuron lesions.

## Bilateral pyramidotomy causes changes in the pattern of excitatory and inhibitory boutons in the spinal cord

One of the more potent anti-spasticity drugs is baclofen, a $GABA_B$ receptor agonist (*Stempien and Tsai, 2000*). Although it may result in adverse systemic side effects and increasing tolerance during long-term treatment, it is evident that increasing inhibition in the spinal cord is an effective treatment for spasticity. Spasticity is associated with a decrease of inhibitory synapses and inhibitory post-synaptic potentials in motor neurons (*Kapitza et al., 2012*; *Kakinohana et al., 2012*; *Boulenguez et al., 2010*) plus concurrent increase of excitatory synapses upon and persistent inward currents in motor neurons (*Tan et al., 2012*; *Toda et al., 2014*; *Bennett et al., 2001*; *Hultborn et al., 2013*). We give evidence that there is an increase of vGluT1 boutons from proprioceptive afferents onto motor neurons and less presynaptic inhibition of proprioceptive afferents in the cervical spinal cord after bilateral pyramidotomy. We also show increased motor neuron excitability through changes in the ion transporter KCC2, but no change in a marker of inhibitory GABAergic synapses/boutons as shown by immunolabeling for vGAT near motor neurons (*Figures 4*, *5*, *Figure 4—figure supplement 1*).

Functionally, knocking out presynaptic GABAergic interneurons results in limb oscillations during directed reaching for sugar pellets (*Fink et al., 2014*). Our bilateral pyramidotomy model results in reduced presynaptic inhibition, oscillatory like movements during the loading response of the stance phase and oscillatory clonic movements of the forelimb which may result from aberrant spinal processing of mechanoceptive and proprioceptive inputs (*Figures 1*, *3*, *Figure 1—figure supplement 2* and *Videos 2* and *3*). These oscillations are also seen in the non-human primates with bilateral pyramidotomy during reaching for food morsels (Supplementary Videos 2, 3 and 8 in [*Lemon et al., 2012*]). Intriguingly, our work indicates that there is not only an increase in excitation (*Figure 4*), but also loss of patterning specificity after bilateral pyramidotomy: We observed abnormal monosynaptic connectivity to radial motor neurons by afferents from an antagonistic flexor nerve (*Figure 4*) and abnormal polysynaptic connectivity to ulnar motor neurons by afferents from the radial nerve (*Figures 1*, *3*). Anatomically, we showed increased numbers of flexor proprioceptive afferent synapses onto motor neurons supplying extensor muscles and neurophysiologically, increased connectivity between flexor and extensor afferents to flexor motor neurons. We propose that this leads functionally to co-contraction or oscillatory muscle activation of flexor and extensor muscles, (including spasms and clonus), which we observed in the open-field testing.

In conclusion, an effective treatment for spasticity should re-balance the excitatory and inhibitory networks in the spinal cord, but also should ensure correct synaptic connectivity between extensor and flexor muscle groups. We have shown that delayed intramuscular delivery of Neurotrophin-3 to flexor muscles can achieve this in adulthood after bilateral pyramidotomy lesion.

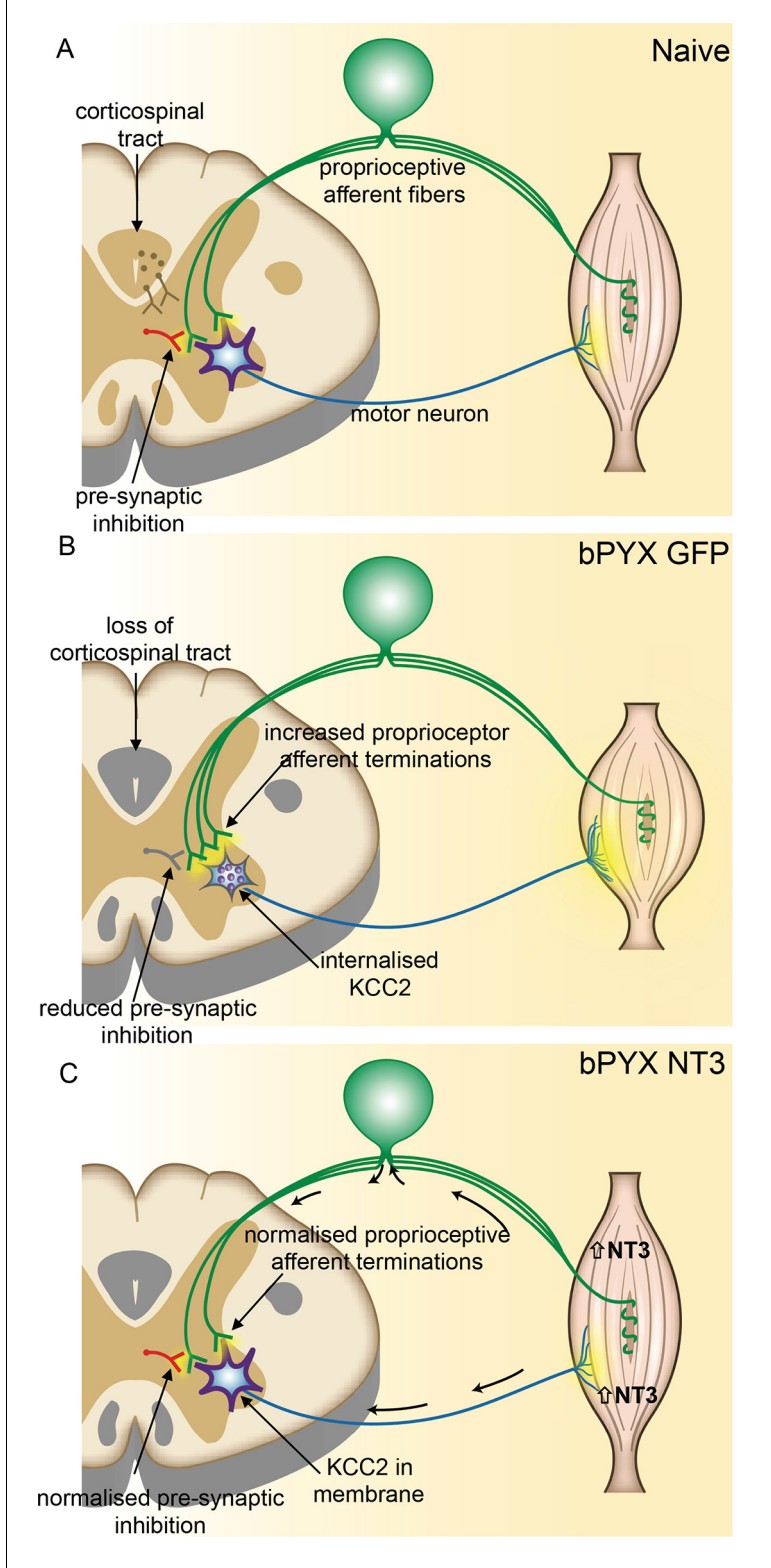

**Figure 6.** Spinal hyper-excitability causing hyperreflexia, spasms and disordered sensorimotor control is normalized by intramuscular Neurotrophin-3 treatment. (**A**) In uninjured healthy conditions, there is a balance between excitatory (afferent and descending) networks, pre-synaptic inhibition and motor neuron excitability. (**B**) After loss of corticospinal innervation, there is a loss of specificity in connections between proprioceptive afferents, spinal interneurons and motor neurons. Indeed, excitatory (afferent) terminations in the spinal cord are increased
*Figure 6 continued on next page*

*Figure 6 continued*

in number and there is a reduction in molecular markers of pre-synaptic inhibition upon proprioceptive afferents. We propose that this causes increased spinal excitability. Moreover, KCC2 is internalized from the membrane of motor neurons rendering them more excitable. Together, these result in increased spinal reflexes, hallmarks of spasticity. (C) Injection of AAV-NT3 into flexor muscles causes trafficking of NT3 to the cervical DRG by retrograde transport (and, possibly, to a lesser extent, by systemic transport). Treatment normalizes the pattern of proprioceptive afferent terminations and restores pre-synaptic inhibition upon proprioceptive afferents. NT3 is also retrogradely transported to motor neurons where KCC2 levels are normalized in the membrane. Multiple spinal reflexes involving treated muscles are normalized. We propose that this is the result of restored specificity of connections between proprioceptive afferents and appropriate motor neurons and normalised motor neuron excitability. Abnormal forelimb movements including spasms are reduced and walking on the ladder is improved. In conclusion, delayed intramuscular treatment with AAV-NT3 results in neurophysiological, molecular and behavioural improvements after CNS injury.

## Importance of proprioceptive afferent feedback

Our findings emphasize a need to re-balance excitatory and inhibitory spinal networks and to increase specificity of afferent connectivity in the spinal cord after CNS injury to treat sensorimotor abnormalities and hyperreflexia. Normally, muscle spindles are involved in mediating this balancing act by giving feedback via proprioceptive neurons. Indeed, mice lacking functional muscle spindles (lacking proprioceptive feedback) show impaired re-organization of spinal circuitry and restricted locomotor recovery after CNS injury (*Takeoka et al., 2014*; *Akay et al., 2014*). Muscle spindles synthesize neurotrophin-3 (*Copray and Brouwer, 1994*) and this may be one of the feedback signals in proprioceptive neurons underlying spontaneous recovery after spinal cord injury.

Bilateral corticospinal tract injury causes spinal cord denervation and subsequently spinal circuitry reorganization. Corticospinal axons and proprioceptive afferents compete for synaptic space in the spinal cord in development and also in adulthood (*Jiang et al., 2016*). Premotor interneurons receive direct input from the corticospinal tract and cutaneous and proprioceptive afferent fibers (*Brownstone and Bui, 2010*); following loss of corticospinal input, proprioceptive afferents may invade the empty synaptic space on these interneurons. The synaptic space on motor neurons might be reorganized more indirectly: bilateral pyramidotomy causes a significant loss of input to spinal interneurons, which may then withdraw some of their input to motor neurons (*i.e.*, anterograde trans-neuronal synaptic loss akin to activity-based synaptic pruning in development) creating more space for sensory afferent synapses. Our data supports this proposition. We have established that after supraspinal injury proprioceptive afferent input reorganized and showed aberrant connection patterns onto motor neurons in the spinal cord (*Figure 6C*). Intramuscular over-expression of Neurotrophin-3 regulated spinal circuitry re-organization and resulted in reduced sensorimotor abnormalities and normalized hyperreflexia.

## Did Neurotrophin-3 reduce spasticity via retrograde signaling?

Motor neurons, low threshold cutaneous mechanoceptive afferents and proprioceptive afferents express trkC (*McMahon et al., 1994*), which is the main receptor for neurotrophin-3. Neurotrophins are retrogradely transported in signaling endosomes (after they bind to and are internalized with their receptors) (*Helgren et al., 1997*; *Harrington et al., 2011*; *Sharma et al., 2010*). We injected an AAV1 expressing neurotrophin-3 into skeletal muscles, which are innervated both by proprioceptive afferents and motor neurons. We found increased levels of neurotrophin-3 protein in the ipsilateral cervical dorsal root ganglia and in motor neurons. We can exclude transport of the AAV1 viral vector as we did not detect the transgene of human Neurotrophin-3 in the ipsilateral cervical DRG with qRTPCR. We also did not detect increased levels of rat neurotrophin-3 mRNA in the ipsilateral cervical dorsal root ganglia after intramuscular injections of an AAV1 expressing the human NT3 transgene as shown by RNA sequencing (Kathe et al, unpublished). We expect that the majority of changes in spinal cord patterning and excitability was mediated via proprioceptive afferents and motor neurons. However, we cannot exclude the possibilities that cutaneous afferents may have modulated spinal reflexes (*Bui et al., 2013*) as it is likely that skin afferents were also exposed to a low concentration of Neurotrophin-3; or that neurotrophin-3 acted to some degree through the

vascular circulation systemically although we didn't detect a significant increase protein in muscles other than the injected ones or in the spinal cord. Notwithstanding our current results, systemic delivery routes of delivery for neurotrophin-3 may be effective when higher doses of the protein are administered, for example by intrathecal infusion after spinal cord injury (*Bradbury et al., 1999*, *1998*). Ongoing work in our laboratory seeks to identify the optimal biopharmaceutical delivery route and form (protein, gene therapy, agonist antibody or small synthetic molecule) for neurotrophin-3 and for targeting its receptors after spinal cord injury and stroke.

## Mechanisms of recovery with neurotrophin-3

Neurotrophin-3 had several effects on spinal networks that likely contributed to normalization of mono- and polysynaptic reflexes and resulted in functional recovery. Firstly, Neurotrophin-3 caused a re-balancing of excitatory and inhibitory networks in the spinal cord. It restored the specificity and number of excitatory boutons from proprioceptive axons onto motor neurons and a marker of pre-synaptic inhibition to normal. Secondly, neurotrophin-3 normalised the levels of KCC2 in ulnar and radial motor neuron membranes to normal. KCC2 subcellular location was modified either after direct retrograde transport of neurotrophin-3 in motor axons (which might mean even the non-significant trend of increased neurotrophin-3 protein in the triceps brachii was sufficient to affect its radial motor neurons), or it was secondary to changes in afferent or supraspinal fiber organization. Finally, neurotrophin-3 also affected plasticity of extrapyramidal supraspinal pathways. We showed that neurotrophin-3 normalized the pattern of serotonergic fibers in the cervical spinal cord. Given that other pathways like rubrospinal tract neurons also express trkC receptors (*Tetzlaff et al., 1994*; *King et al., 1999*) it is possible that these pathways also undergo sprouting within the cervical spinal cord and may contribute to functional recovery (*Raineteau et al., 2002*).

Taken together, we propose that all these anatomical effects of neurotrophin-3 normalize spinal reflexes and reduce spasticity that can interfere with functional recovery. This improves locomotor function. Motor function and restoration of normal reflexes are interdependent and mutually reinforcing as it has been shown that neurorehabilitative training ameliorates reflexes (*Côté et al., 2011*) and reflex training ameliorates neurorehabilitation (*Thompson and Wolpaw, 2015*).

## Neurotrophin-3 is a translationally attractive therapy for treating spasticity as part of a movement disorder

Spasticity seldom occurs in isolation, but more commonly as part of larger syndromes such as upper motor neuron syndrome or the pyramidal syndrome associated with stroke and spinal cord injury. These syndromes often affect other aspects of movements and sensory processing including locomotion, fine motor function or cutaneous sensation. Existing anti-spastic treatments, which reduce exaggerated reflexes and muscle tone, often do not improve the movement disorder (*Dietz and Sinkjaer, 2007*; *Corry et al., 1997*; *Latash and Penn, 1996*). In contrast, we have shown that neurotrophin-3 has the added benefit of improving other aspects of motor and sensory function after CNS injury. Here, we have shown intramuscular treatment with neurotrophin-3 not only normalized reflexes, but also improved locomotion and grip strength. Moreover, neurotrophin-3 treatment did not cause detectable pain (mechanical allodynia). In previous studies, our group used delayed, intramuscular neurotrophin-3 to treat stroke and observed improved fine motor function and tactile sensation in adult and elderly rats (*Duricki et al., 2016*).

Importantly, peripheral administration of neurotrophin-3 protein is safe and well-tolerated in Phase 2 human clinical trials even with high doses (*Chaudhry et al., 2000*; *Coulie et al., 2000*; *Sahenk et al., 2005*; *Parkman et al., 2003*; *Sahenk, 2007*). Our findings suggest that intramuscular neurotrophin-3 reduces signs of spasticity and improves functional recovery by regulating spinal connectivity after supraspinal injury. Moreover, a gene therapy involving intramuscular injection of an AAV1 encoding human neurotrophin-3 under the control of a muscle-specific promoter, which was successfully tested in a mouse model of Charcot Marie Tooth neuropathy (*Sahenk et al., 2014*), now has FDA approval for a clinical trial for patients with Charcot Marie Tooth 1A (Professor Zarife Sahenk, personal communication, August 2016). Taken together, this paves the way for neurotrophin-3 as a therapy for CNS injury.

## Materials and methods

### Animals

All animal procedures were conducted in accordance with the UK Home Office guidelines and the Animals (Scientific Procedures) Act of 1986. Animals were housed in groups in standard housing conditions on a 12 hr light/dark cycle with food and water ad libitum. 60 female Lister hooded rats (Charles River; outbred; 190–250 g) were used. Generally, the group size was 10 or more rats per treatment in line with previous spinal cord injury/ stroke in vivo studies (*Duricki et al., 2016*; *Kathe et al., 2014*). All experiments were performed in a randomized block design and blinding during experiments and analysis was achieved by using viral vector aliquots coded by a third party. Codes were broken at the end of the studies. Materials and methods and results have been written in accordance with the ARRIVE guidelines for publishing in vivo research.

### Pyramidotomy lesions

As described in detail previously (*Kathe et al., 2014*), rats were anaesthetized with 5% isoflurane in 1 L/min oxygen. The ventral neck region was shaved and swabbed with alcohol wipes. A 2 cm long midline incision was made. Overlying tissue was blunt dissected until the trachea was exposed. The trachea was displaced to the side and underlying muscle blunt dissected until the ventral side of the basioccipital bone was exposed. Next, a hole was drilled. Both pyramids were cut using Vannas microscissors and cuts were retraced with a 26-G needle. Any bleeding was stopped and the skin was sutured. Carprieve was given as the analgesia during surgery and for the following 2 days (5 mg/kg, subcutaneous, twice daily).

### Adeno-associated viral vectors

Details of the adeno-associated viral vectors (AAV) used in these studies have been described previously (*Duricki et al., 2016*). In short, we used AAV plasmids encoding either human prepro-Neurotrophin-3 or GFP as a control treatment. The Viral Vector Core of University of Pennsylvania packaged viral vectors into AAV1 capsids. Viruses were titer-matched for all experiments.

### Muscle injections for pyramidotomy study

Muscle injections were made into the forelimb flexor muscles on the previously dominant side. $3 \times 10^{10}$ viral genomes of an AAV1 encoding human prepro-Neurotrophin-3 or AAV1-GFP in 90 µl PBS containing 5% sucrose were injected with Hamilton syringes bearing 31-G needles into the dominant forelimb at 24 hr after pyramidotomy. Rats were anaesthetized with isoflurane. The forelimb was shaved and swabbed with alcohol wipes. An incision above the biceps brachii was made and six volumes of 5 µl were injected. Another incision was made parallel to the ulnar bone. In total, eight volumes of 5 µl were injected into following muscle groups: flexor carpi radialis, flexor digitorum profundus, flexor carpi ulnaris, palmaris longus, flexor digitorum sublimis and pronator teres. The skin was sutured. One 10 µl injection was made through the skin into the abductor digiti quinti and one 10 µl injection was made into the plantar pad. Rats received peri-surgical Carprieve.

### Muscle injections for biodistribution study

Cervical DRGs from the pyramidotomy study were used for RNA sequencing or immunohistochemistry (unpublished results), muscles were used for ELISA and spinal cords were used for histology (data shown). Accordingly, a second cohort of rats were generated for evaluating the biodistribution of NT-3 in DRG and other tissues. Rats were anaesthetized with isoflurane as previously described. Muscle injections were aimed at the largest muscle groups in the forelimb. $3 \times 10^{10}$ viral genomes in 90 µl PBS containing 5% sucrose were injected with Hamilton syringes bearing 31-G needles into the triceps brachii and biceps brachii on the left side. The forelimb was shaved and swabbed with alcohol wipes. An incision above the biceps brachii was made and six volumes of 5 µl were injected. The skin was sutured. Another incision was made above the triceps brachii. Twelve volumes of 5 µl were injected into the lateral and long heads of the triceps. The skin was sutured. Animals received pre-surgical analgesia. They were sacrificed at either 4 days or 4 weeks post-surgery.

## Tracing

Three days prior to terminal electrophysiology experiments, we retrogradely traced ulnar (flexor) and radial (extensor) motor neurons. Rats were anaesthetized and the axilla on the treated side was shaved and swabbed with alcohol. A cut was made over the lateral end of the pectoralis major. Skin and overlying muscle were blunt dissected until the ulnar, median and radial nerves were visible. 2 μl of 1% Cholera Toxin beta subunit (CTb, List Biological Laboratories) in 0.9% saline and 2 μl of 1% Fast Blue (Sigma) in 0.9% saline (Fresenius Kabi, Cheshire, UK) were injected with a 34-G needle on a Hamilton syringe into the ulnar and radial nerves respectively. Overlying skin was sutured and analgesia was given.

## Behavioural testing

Rats were trained for 3 weeks pre-surgery and then assessed fortnightly post-surgery. Based on the Montoya staircase pellet-reaching test the dominant forepaw was determined prior to pyramidotomy. All behavioural testing was done by an observer blinded to treatment group.

## Montoya staircase pellet reaching test

Rats were initially food-restricted to 15 g per rat overnight during the pre-training phase, but not during testing. The staircase was filled with 3 sucrose pellets (45 mg) into each well of the 7 stairs on either side, totaling 21 pellets per side. Rats were left for 15 min in the Montoya staircase. The total number of eaten and displaced sucrose pellets was counted for each forelimb separately.

## Horizontal ladder test

Rats crossed a 1 m long horizontal ladder with randomly spaced rungs three times. Videos of the runs were analysed at a later time-point and errors were counted for the affected forelimb. Error values are represented as a percentage of total steps per limb.

## Grip strength test

Grip strength was assessed with a bilateral grip strength device (Linton) which measures grip strength of each arm separately and simultaneously. The grip strength device measures the force with which rats held onto the two bars (each containing an independent force transducer) while being gently pulled away. During each fortnightly testing session, rats were tested 4 times. The four values for each limb were averaged to give the test scores per animal at each time.

## Automated von Frey test

Rats were placed in Plexiglas boxes with a gridded floor. They were left to settle for at least 30 min before testing started. Nociceptive responses to mechanical stimulation of the plantar surface of the forepaws was assessed with the automated dynamic plantar aesthesiometer (37400–001, Ugo Basile), max force 50 g, ramp duration 20 s. Each forepaw was tested separately 3 times at each time-point. The values were averaged to give the test scores per rat at each time point.

## Open-field test: scoring for spasticity and disordered sensorimotor control of the forelimb

Rats were placed into a 50 cm diameter Plexiglas cylinder and videotaped for 3 min fortnightly. Videos were scored by a blinded observer using the Scoring sheet shown in *Figure 1—figure supplement 2*.

Examples of each sign of spasticity and sensorimotor abnormality that was scored are shown in *Video 2*. For purposes of comparison, *Video 1* shows a normal rat exhibiting none of these signs. Each forelimb was scored separately, and scores were added up to give a total score per rat (yielding a maximum of 24). Other signs of spasticity (especially those relating to the hindlimb and tail) were not scored (*Video 3*). The following signs of spasticity or sensorimotor abnormality were scored, using the operational definitions given:

1. Forepaw digit flexion during swing phase. A score of 1 was given if the forepaw digits were held in a fully flexed position during the swing phase. A score of 0 was given if the forepaw digits appeared flaccid or atonic (i.e., neither flexed nor extended). A score of 0 was given if

the forepaw digits were extended. The score was based on the digit position most frequently observed during the three minutes.

2. Joint movements during swing phase. Movements of the wrist, elbow and shoulder were assessed during the swing phase. A score was given to each joint (with a maximum score of 2 per joint). 1 point was given if the joint appeared rigid or showed corrective movement (alternating trajectories in the mid-swing phase) and 2 points were given if the joint appeared rigid and showed corrective movements. A score of 0 was given if the joint movement was smooth or flaccid. The score was based on the type of movement frequently observed for each joint during the three minutes. The maximum score a single forelimb could reach is six (i.e., 3 joints × 2 points).

3. Stance width: A 'Normal stance' was defined as how the forepaw was placed by normal rats (i.e., slightly medial to shoulders) and was given a score of 0. A 'Wide stance' was defined as when the forepaw was placed more laterally than that seen in normal rats. A wide stance is often seen in rats with neurological injury because this compensatory strategy improves stability. A score of zero was given for a wide stance because it does not indicate increased tone of flexor muscles. A 'Narrow stance' was defined as when the forepaw was placed more medially than that seen in normal rats, with forepaws aligned rostrocaudally or crossed. A narrow stance was given a score of one because it indicates increased muscle tone of flexor muscles. The score was based on the type of stance most frequently observed during the three minutes.

4. Forelimb movements during the loading response of the stance phase. A score of 0 was given for a normal loading response during the stance phase. A score of 1 was given for 'Repeated muscle jerks during onset of stance'. This was defined as when, during the loading response of the stance phase, the forelimb made contact but then withdrew immediately (with one or more cycles of this 'bounce'). A score of 0 was given if the plantar surface of the forepaw was not placed down during the loading response of the stance phase ('No plantar stance'), i.e. dorsal stepping. The score was based on which of these was observed most frequently during the three minutes.

5. Other signs of sensorimotor abnormality: prolonged muscle contractions, single and repeated muscle jerks. A prolonged muscle contraction was defined as a longer than 2 s 'on' muscle activation (e.g., flexor or extensor). A single muscle jerk was defined as a rapid, single 'on-off' muscle activation. A repeated muscle jerk was defined as a rapid and repeated 'on-off' muscle activation (other than in the loading response of the stance phase, to avoid double counting). A score of 1 was given for each of these behaviours if they were observed at least once.

Many of these signs of sensorimotor abnormalities have been observed in non-human primates (macaque monkeys) after bilateral pyramidotomy, including (1) narrower stance of forelimbs during quadrupedal walking (Supp Video 6 of (*Lemon et al., 2012*), (2) dampened oscillatory movements during forelimb movements (Supp Video 2 and 8 of [*Lemon et al., 2012*]) and (3) prolonged flexor muscle contractions in sitting positions (Supp Video 7 of [*Lemon et al., 2012*]).

## Hoffmann-reflex testing

The H-reflex was assessed at baseline and every two weeks post-surgery. Rats were anaesthetized with 30 mg/kg ketamine and 0.1 mg/kg medetomidine. Two 24-G needle electrodes were inserted across the medial plantar side of the wrist to stimulate the ulnar nerve (via a constant current isolated pulse stimulator, stimulus width 100 µs, Neurolog, Digitimer). Two recording electrodes were inserted into the abductor digiti quinti to record electromyograms. The signal was amplified (4000-fold), filtered (with a pass band of 300 Hz to 6 kHz), digitized via PowerLab, visualized and analysed with LabChart. The M-wave is evoked by excitation of motor axons. The H-wave is a short latency reflex which includes monosynaptic connections: Ia proprioceptive afferents synaptically activate motor neurons in the spinal cord (*Figure 1J*). The threshold (1X) was determined as the lowest stimulation intensity that elicited an H-wave response in at least 75% of the recordings. First, we tested the responses to increasing stimulus intensities at 0.1 Hz up to 2x Threshold, which activates low threshold afferent fibers. M-wave and H-wave amplitudes were normalized to the maximum M-wave that we recorded at higher stimulation intensities (up to a maximum threshold of 2x threshold). Next, we tested the frequency dependent depression of the H-wave. This is also commonly referred to as post-activation depression or as rate-dependent depression. We stimulated every 10 s with paired stimuli at inter-stimulus intervals from 10 s to 0.1 s. The H-wave amplitude of the test stimulus was normalized to the H-wave amplitude of the conditioning stimulus. 25 paired stimuli per frequency were averaged and plotted as a frequency-depression curve. The area under the curve was

calculated to give an H-wave measure of hyperreflexia (*Figure 1—figure supplement 3F*). Post-stimulation, rats were given atipamezole (Antisedan, 2 mg/ml) and monitored until fully awake.

## Cervical dorsal root axotomy

Three naïve animals were used to confirm the H-reflex set-up. Animals were stimulated and recorded from as stated in H-reflex methods. Immediately after the recording, six cervical dorsal roots were cut on the stimulation/recording side. A dorsal midline incision was made above the cervical vertebrae. Muscle tissue was blunt dissected. Laminectomies were performed from C3 to C8. The dura was opened and dorsal roots C3 to C8 were cut with micro-scissors unilaterally. Muscle tissue was sutured in layers and the skin was sutured. Post-surgery animals were kept anaesthetized and the H-reflex recording protocol was repeated. After H-reflex recording, animals were sacrificed by cervical dislocation.

## Terminal electrophysiology involving median, ulnar and radial nerves

Rats were anaesthetized with an intraperitoneal injection of 30 mg/kg ketamine and 0.1 mg/kg medetomidine. A skin cut overlying the pectoralis major from the axilla to the elbow was made and a mineral oil pool was established with the skin flaps. The brachial plexus was exposed and connective tissue was removed from the ulnar, median and radial nerves, which were cut distally. We performed whole nerve recordings from the ulnar nerve, which was mounted on silver-wire hook electrodes (*Bosch et al., 2012*). The median and radial nerves were stimulated to elicit a synergistic and antagonistic response respectively (100 μs, 0.1 Hz, 0–400 μA). Stimulation intensities were chosen based on the previous observations in our laboratory (*Bosch et al., 2012*). Alternatively, the radial nerve was recorded from whilst stimulating the ulnar nerve. The signal was amplified (4000-fold), filtered (with a pass band of 600 Hz to 3 kHz), digitized via PowerLab, visualized and analysed with LabChart. Filtered traces were analysed by measuring the absolute integral (area under the rectified curve) from 1.5 ms to 3.5 ms for monosynaptic responses and 3.5 ms to 12 ms for polysynaptic responses. Values were averaged across 10 recordings per stimulation intensity per animal.

## Tissue preparation

Immediately after the nerve prep, rats were perfused transcardially with PBS pH 7.4 and tissues were dissected rapidly. Tissue for sectioning on the cryostat was post-fixed by immersion in 4% paraformaldehyde in PBS pH 7.4 overnight and cryoprotected in 30% sucrose in PBS pH 7.4. Tissue was frozen and embedded in O.C.T., spinal cords and DRGs were sectioned transversely at 30 μm or 10 μm thickness respectively. Tissues for protein analysis and for RNA extraction were snap-frozen in liquid nitrogen after dissection and then stored in a −80°C freezer. Detailed protocols for histochemistry, immunohistochemistry, protein analysis, RNA extraction, and qRTPCR and analysis can be found below.

## Histochemistry and immunohistochemistry

Eriochrome cyanine staining was performed as described previously (*James et al., 2011*) using 3 sections per rat. In summary, slides were immersed in PBS pH 7.4 for 5 min, then dehydrated in a graded ethanol series for 5 min each. Sections were cleared in Histochoice or Xylene for 15 min before rehydrated in a graded ethanol series. Slides were washed for 5 min in distilled water before stained in 0.16% Eriochrome Cyanine R (Sigma-Aldrich, Fluka 32752), 0.5% sulphuric acid (Sigma 84728) and 0.4% iron chloride (Alfa Aesar 12357) in $dH_2O$ for 10 min to visualize myelin. Slides were washed in distilled water twice for 5 min and then differentiated in 0.5% aqueous ammonium hydroxide until desired staining intensity is reached, but no more than 2 min. Slides were washed twice for 5 min in distilled water and then dehydrated in a graded ethanol series. Finally, slides were left in Histochoice for 10 min or longer before cover-slipping with DPX.

Immunohistochemistry was performed with following standard protocol: Tissue was blocked in 10% bovine albumin serum (Sigma, A3059) in PBS for 1 hr, then incubated with the relevant primary antibody (Antibody Table) overnight at room temperature. After 4 washes with PBS, the appropriate secondary antibody (Antibody Table) was applied for 90 min at room temperature. Images were taken with a Zeiss Imager Z.1 fitted with an AxioCamMRm or with an LSM 710 microscope (Carl

Zeiss, 0.3 µm thick optical sections). Generally, no GFP expression was observed in unstained sections of DRG or spinal cord.

All antibodies were verified by their manufacturer (Antibody Table). The serotonin antibody was raised in rabbit and specific staining is inhibited by pre-incubation of the diluted antiserum with serotonin or serotonin-BSA (manufacturer's specification). The vGAT antibody was raised against amino acids 75–87 of rat vGAT and its specificity for mammalian vGAT was verified by demonstrating loss of staining using KO mice by the manufacturer. The vGluT1 antibody was raised against amino acids 456–560 of rat vGluT1. Its specificity for mammalian vGluT1 was verified by demonstrating loss of staining in KO mice by the manufacturer. The KCC2 antibody recognizes the residues 932–1043 of the rat ortholog and is routinely evaluated by western blot in rat brain membrane preparations by the manufacturer. The Neurotrophin-3 antibody is a synthetic peptide corresponding to the human prepro-Neurotrophin-3 amino acids 174- 189 (the epitope is part of the mature peptide sequence). Staining can be inhibited with Neurotrophin-3 peptide (manufacturer's specification).

| Antibody | Supplier and cat number | Concentration |
| --- | --- | --- |
| Rabbit anti-serotonin | Sigma, S5545 | 1:6000 |
| Mouse anti-vGAT | Synaptic Systems 131 011 | 1:200 (+0.1%TX-100) |
| Rabbit anti-vGluT1 | Synaptic Systems 135 302 | 1:1000 (+0.1%TX-100) |
| Goat- anti-CTb | List Laboratories 703 | 1:2000 |
| Rabbit anti-KCC2 | Millipore 07-432 | 1:500 |
| Rabbit anti-NT3 | Abcam 65804 | 1:500 |
| Alexa 488 donkey anti-mouse | Life Technologies A21202 | 1:1000 |
| Alexa 488 donkey anti-rabbit | Life Technologies A21206 | 1:1000 |
| Alexa 594 goat anti-rabbit | Life Technologies A11012 | 1:1000 |
| Alexa 488 donkey anti-goat | Life Technologies A11055 | 1:1000 |
| DyLight 650 donkey anti-goat | Abcam ab96938 | 1:1000 |

Antibody table: Table showing primary and secondary antibodies.

## Image analysis

Image analysis was performed with Image J or Zen Imaging software.

### Analysis of serotonin immunohistochemistry

Images were acquired with a Zeiss Imager Z.1 fitted with an AxioCamMRm. Pixel intensities were measured for each animal individually, normalized to an unstained area in the white matter and then statistically analysed (*Grider et al., 2006*). The following areas were measured for serotonin immunohistochemistry: Laminae I-III and the ventral horn (laminae VII to IX) on the affected side. For each area, three C8 sections per animal were analysed and then averaged.

### Analysis of KCC2 ion symporter levels in ulnar and radial motor neuron membranes

Retrogradely traced ulnar and radial motor neurons were analysed separately with immunohistochemistry with an antibody against KCC2. C7/8 spinal sections were analysed. Once motor neurons were identified, 3 cross-sectional lines were drawn across the motor neurons and pixel intensities were measured along the lines. The area under the curve, where the lines cross the membrane, was measured. Thus, 6 values for each motor neuron were measured, which were averaged before further analysis. 25 motor neurons were analysed per rat and per motor neuron type.

### Analysis of synaptic boutons by confocal fluorescence microscopy

To determine the number of boutons in close proximity to motor neurons or other boutons, consecutive confocal images (0.3 µm thick optical sections) were acquired as z-stacks (total average thickness: 15 µm) using a 63x objective. Z-stacks were analysed by scrolling through the different optical

sections. To identify inhibitory boutons in close proximity to terminals, immunohistochemistry was performed with antibodies for a pre-synaptic marker for GABAergic synapses (vGAT+) and a pre-synaptic marker labeling proprioceptive or cutaneous sensory afferents terminals (vGluT1+). 250 vGluT1 boutons were scored in the ventral horn for each animal by a single and blinded experimenter: either the bouton was in very close proximity to at least one vGAT bouton (score = 1) (e.g., identified by yellow pixels in 15 µm z-stacks when projected into a single plane; see *Figure 4H,J*) or not in close proximity to any vGAT boutons (score = 0) (see *Figure 4I*). To identify boutons in close proximity to motor neurons, vGluT1 and vGAT boutons were counted on at least 25 motor neurons per animal, which had been retrogradely traced with CTb or Fast Blue (see above). The average number of vGluT1 boutons in close proximity to ulnar and radial motor neurons was calculated separately. In each case the mean score was calculated for each rat.

## Enzyme-linked-immunosorbent assay

Fresh tissue samples were immediately snap-frozen in liquid nitrogen and then stored in a −80°C freezer. Protein was extracted with RIPA buffer (50 mM Tris HCl pH 7.5, 150 mM NaCl, 2 mM EDTA, 1% Triton X-100, 0.1% SDS) and a hand-held mechanical homogenizer. After 1 hr incubation time, samples were spun at 17,900 g at 4°C for 15 min. The supernatant was collected and stored at −20°C. Blood samples were allowed to clot at room temperature for up to 1 hr and then centrifuged at 17,900 g at 4°C for 15 min. The serum was collected and stored at −20°C. ELISAs were performed with the Human NT-3 DuoSet kit (DY267, R&D systems) according to manufacturer's instructions with some modifications. In summary, plates were pre-coated with the capture antibody at room temperature overnight. Plates were washed three times with the wash buffer and then blocked with the reagent buffer for one hour. The washing step was repeated and 100 µl samples were applied undiluted in duplicates. Plates were left to incubate on a shaker at room temperature for 5 hr. Post-incubation, plates were washed and the detection antibody was applied for overnight at 4°C. The next day, plates were washed and streptavidin was added to each well for 20 min. The washing step was repeated and the substrate solution applied for 20 min. The reaction was stopped with 1N $H_2SO_4$ after which the optical density of the plate was read with a micro-plate reader at 540 nm and 450 nm (Spectramax 340PC). Concentrations were calculated with the help of a standard curve on a four-parameter logistic curve fit. Neurotrophin-3 concentrations were normalized to the total protein amount determined with the Bicinchoninic Assay (Millipore 71285–3) and then expressed relative to naïve control levels.

## Total RNA extraction and qRTPCR

Total RNA was extracted from the C7 DRG from the rats according to the manufacturer's instructions (Qiagen, 74104). Samples were also DNase I-treated (Qiagen, 79254). 110 ng of total RNA per sample was used for reverse transcription to obtain the cDNA (Invitrogen, 18080–0440–044) using random primers. We used following primer pairs for quantitative real-time PCR which was performed in triplicates (Roche, LightCycler 480 II): human Neurotrophin-3 forward: 5' -GAA-ACG-CGA-TGT-AAG-GAA-GC-3'; human Neurotrophin-3 reverse: 5'- CCA-GCC-CAC-GAG-TTT-ATT-GT-3'; GAPDH forward: 5'-ATG-GGA-AGC-TGG-TCA-TCA-AC-3'; GAPDH reverse: 5'-CCA-CAG-TCT-TCT-GAG-TGG-CA-3'. A standard curve for human Neurotrophin-3 was obtained using threefold dilutions of human fetal brain cDNA (Stratagene) applied in duplicate/triplicate. A standard curve for rat GAPDH was obtained using threefold dilutions of cDNA derived from a pool of embryonic day 16–18 rat brains. NT-3 concentrations were normalized to GAPDH levels.

## Statistics

All results are expressed as mean ± Standard Error of Mean (SEM). All data points were analysed with appropriate parametric tests as stated in figure legends. Outliers were included. The experimental unit (n) was the rat. Generally, behavioural tests and frequency-dependent depression were analysed by comparing between groups using two-way repeated measures ANOVA (group or group*time interactions are given in Figure Legends). In the Figure Legends to *Figure 1* and *Figure 3* statistical analyses are based on all groups (i.e., naïve, bPYX GFP and bPYX NT3). Eriochrome cyanine and ELISA data were analysed by comparing between groups (naïve, bPYX GFP or bPYX NT3) and sides (ipsilateral vs contralateral) using two way ANOVA (group or

group*side interactions are given in Figure Legends). ELISA data for liver were analysed by an unpaired t-test. The threshold for significance was p<0.05. Statistical analysis was performed in SPSS v.19 or GraphPad Prism v.6.

### Ethical approval

All animal work was done in accordance with the United Kingdom Animals (Scientific Procedures) Act of 1986 and was approved by the Animal Welfare and Ethical Review Body (AWERB) of King's College London. The work was conducted under Home Office Project License number 70/7865.

## Acknowledgements

We thank Prof. M Ramer and Emeritus Prof. T Sears for help with setting up the forelimb H-reflex testing; Dr. HD Shine and Dr. Q Chen for advice on ELISAs; Prof. F Gage (Salk Institute) for providing us with the AAV-NT3 and AAV-GFP plasmids, the University of Pennsylvania Vector Core for producing the viral vectors, and Dr. T Lieberthal for creating the schematic in *Figure 6*. We thank Dr. S Ashford, Dr. J Taylor and Dr. E Bravo Esteban for advice regarding signs of spasticity in rats. The research leading to these results has received funding from the International Spinal Research Trust's Nathalie Rose Barr Studentship and the Rosetrees Trust, a Serendipity grant from the Dunhill Medical Trust (SA21/0512), the European Research Council under the European Union's Seventh Framework Programme (FP/2007-2013) / ERC Grant Agreement n. 309731, and King's College London Graduate Teaching Assistant Program.

## Additional information

### Funding

| Funder | Grant reference number | Author |
| --- | --- | --- |
| Spinal Research | Nathalie Rose Barr studentship | Claudia Kathe<br>Lawrence David Falcon Moon |
| Rosetrees Trust | | Claudia Kathe<br>Lawrence David Falcon Moon |
| Dunhill Medical Trust | Serendipity Grant (SA21/0512) | Thomas Haynes Hutson<br>Lawrence David Falcon Moon |
| European Research Council | ERC grant agreement: 309731 | Lawrence David Falcon Moon |
| Kings College London | Graduate Teaching Assistant Program | Claudia Kathe<br>Lawrence David Falcon Moon |

The funders had no role in study design, data collection and interpretation, or the decision to submit the work for publication.

### Author contributions

CK, Conception and design, Acquisition of data, Analysis and interpretation of data, Drafting or revising the article; THH, Acquisition of data, Analysis and interpretation of data; SBM, Supervision, Drafting or revising the article; LDFM, Conception and design, Analysis and interpretation of data, Drafting or revising the article

### Author ORCIDs

Claudia Kathe, http://orcid.org/0000-0001-6441-1755
Lawrence David Falcon Moon, http://orcid.org/0000-0001-9622-0312

### Ethics

Animal experimentation: All animal work was done in accordance with UK Home Office guidelines and the United Kingdom Animals (Scientific Procedures) Act of 1986 and was approved by the Animal Welfare and Ethical Review Body (AWERB) of King's College London.

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
