## [Decision Letter]

Thank you for submitting your article "Intramuscular neurotrophin-3 normalizes reflexes and improves mobility after CNS injury in rats" for consideration by *eLife*. Your article has been reviewed by three peer reviewers, and the evaluation has been overseen by a Reviewing Editor and Eve Marder as the Senior Editor.

There was a general enthusiasm for your study, which introduces a new rodent model that captures aspects of spasticity after injury of the corticospinal tract and the potential positive effect of NT3 on the aberrant reflexes and motor disabilities. However, several major concerns about the presentation of the model, the electrophysiological and molecular data, and the mechanisms for the NT3 effect, as well as a number of other issues were raised. To alleviate these problems a major revision of the text and better presentation of data as well as elimination of data from the present manuscript are needed, as outlined below. The reviewers have discussed the reviews with one another and the Reviewing Editor has drafted this decision to help you prepare a revised submission.

Summary of the work:

In this manuscript, Kathe et al. describe a new injury rat model with bilateral pyramidotomy which leads to permanent motor impairment of fore and hindlimbs and changes in spinal reflexes resembling those following stroke or partial spinal cord injury. The behavioral changes were observed during spontaneous movements while reflex changes were tested in anaesthetized animals. The authors also show changes in synaptic connectivity onto motor neurons and that molecular factors are up- or down-regulated after in the dorsal root ganglia. Interestingly, many of the reflex changes and the changes in synaptic connectivity appear to be normalized by intramuscular over-expression of NT3 using viral vectors in single muscles, while other aberrant motor behaviors are partially improved. The study presents a new model of CNS injury leading to reflex changes and spasms included in the motor symptoms often described as spasticity in humans after spinal cord injury. The study clearly shows an effect of NT3 treatment on these symptoms suggesting that it may be possible to rectify injury released motor symptoms with this therapy.

Essential revisions:

1) The authors introduce a new model of 'spasticity'. The model and the motor changes and reflex changes that follow the injury need to be presented much better. Essentially Figure 1 should show the anatomical lesion, the observed behavioral motor disorders (spasms, changing in stepping overcrossing, staircase, open field score, etc.), and the essential reflex changes which include the changes in H-reflex, polysynaptic reflexes, and postsynaptic depression in control and in weeks after the injury. In this way the reader will be able to compare this model with previously published models (partial or full SC injury or stroke models). For the reflexes it needs to be made clear what kind of polysynaptic reflexes are tested. The current description is superficial and not convincing. All the stimulated nerves contain skin afferents (in the group I range) that could contribute to polysynaptic reflexes. But there is no mention of this in the text. In addition, the reflexes are difficult to understand even for a spinal motor physiologist because the small diagrams are misleading, and the text and figure legends are insufficient (for example Figure 2C of postsynaptic depression is incomprehensible, and it is not clear that the diagrams in Figure 2G, J are correct as well as the one in Figure 2—figure supplement 1F and K). Please restrict the reflex description to the well-known H-reflex and postsynaptic depression.

2) Having introduced the model and its changes, the NT3 effects should be described in a separate text and figure with the essential reflexes and motor behavior summarized. In this way the authors can avoid jumping back and forth in the description of the model, the electrophysiological findings, and the behavioral characterizations and how these parameters are relevant for spasticity and the treatment with NT3 injections.

3) The definition of spasticity should be clear. However, the discussion of what spasticity really is does not seem very appropriate. There is no right or wrong answer to that (and no "controversy", even if there may be strong opinions). Clearly definitions have to be somewhat different in the clinic and in animal experiments, even if one tries to find animal models that are as close to the clinical situation as possible. It should be clearly stated that selective pyramidal lesion is a rare clinical event and stroke or spinal cord injury related spasticity is much more common.

4) There are generally a number of overstatements that NT3 treatment completely alleviates spasticity, which is not the case. This should be rectified. Also it should be noted that other treatments have been introduced in addition to baclofen.

5) Synaptic analysis is non-trivial and the authors state that they counted a synapse if vGAT and vGlut1 were in 'very close apposition'. What exactly do they mean by this – how close does this need to be to be classed as a synapse and without deconvolution do they think they are resolving individual synapses?

6) How were the 5-HT changes assessed (it is not obvious from the figure that the changes are representative). Needs discussion and comments on how the quantification was done.

7) The NT3 treatment is not easy to understand. What does it do, and how can it act non-retrogradely – is it because the main effect is systemic? Otherwise it is hard to understand that so many symptoms can be affected by a single or few muscle injections. The authors should explain the large tendency in NT3 increase in triceps after virus injection in flexor muscles, and the recovery in radial MNs Figure 4D and in TrkC neurons in Figure 7B and generally discuss the systemic effects.

8) The explanation of the NT3 increase in DRG by its retrograde transport is not supported by the data. Retrograde transport of the active signaling from the nerve endings to the soma is the basis of the long-range actions of the neurotrophins.

9) The PCA analysis does not provide much information, nor does it strengthen the data because it relies on author observations/analysis that have been arbitrarily chosen. Should be deleted.

10) The RNA-seq data should be presented in a comprehensive way with the genes in the main figure. However, since RNAseq has been performed on entire DRGs, it is hard to draw any conclusions from the data. Other neuron types express TrkC in the DRG and are likely to respond to NT3 overexpression. It is therefore not easy to understand the focus on sema 4. It is interesting but there is not proof or even slight indication that this molecule is involved in any of the changes. To show this would require a KO. We think it unreasonable to ask for this for this study, but we think the data are too preliminary at this point to include as an explanation and propose that the entire RNA-seq story be deleted from the paper and saved for a separate publication where the functional role of individual molecules can be addressed.

11) Discussion: The comments on Lawrence and Kuypers' experiments/films are important since the present model is a bilateral lesion of the pyramidal tract! When it comes to the mechanisms (subsection “Underlying mechanisms and treatment choice”) it is not correct that most GABAB receptors are responsible for presynaptic inhibition of Ia afferents. It is also striking that the KCC2 is the only mechanism proposed for cellular mechanisms for the hyperexcitability of motoneurons. What about the persistent inward currents – that were first described by Schwindt and Crill, then by Hounsgaard, Hultborn, Kiehn et al., and Dave Bennett et al. (the 2 latter groups also proving that they are increased following chronic spinal cord lesion, and certainly contribute to spasticity)? This is complemented by experiments by several other groups in Seattle, Chicago and Halifax. This paper is not addressing this question, but a couple of sentences in the Discussion should be a minimum. Finally, how can removal of the CST lead to change in competition onto motor neurons. CSTs do not reach MNs in rats?

12) Figure 8 should be deleted since it there is not support for only retrograde NT3 transport as the mechanism.

13) Title: The title needs to be changed to reflect the revision. In particular it needs to be stated what the CNS injury is and what reflexes that are normalized.

---

## [Author Response]

Essential revisions:

*1) The authors introduce a new model of 'spasticity'. The model and the motor changes and reflex changes that follow the injury need to be presented much better. Essentially Figure 1 should show the anatomical lesion, the observed behavioral motor disorders (spasms, changing in stepping overcrossing, staircase, open field score, etc.), and the essential reflex changes which include the changes in H-reflex, polysynaptic reflexes, and postsynaptic depression in control and in weeks after the injury. In this way the reader will be able to compare this model with previously published models (partial or full SC injury or stroke models). For the reflexes it needs to be made clear what kind of polysynaptic reflexes are tested. The current description is superficial and not convincing. All the stimulated nerves contain skin afferents (in the group I range) that could contribute to polysynaptic reflexes. But there is no mention of this in the text. In addition, the reflexes are difficult to understand even for a spinal motor physiologist because the small diagrams are misleading, and the text and figure legends are insufficient (for example Figure 2C of postsynaptic depression is incomprehensible, and it is not clear that the diagrams in Figure 2G, J are correct as well as the one in Figure 2—figure supplement 1F and K ). Please restrict the reflex description to the well-known H-reflex and postsynaptic depression.*

We have entirely revised Figure 1 in line with your suggestions. We have included lesion assessment, observed behavioural deficits and the essential reflex changes in Figure 1. Reflexes, which were assessed with neurophysiology, include the H-reflex and polysynaptic reflexes in Figure 1J-L and 1M-U , respectively. “Figure 1—figure supplement 3” shows additional H-wave depression data in uninjured naives and at various time-points after the injury including new data for earlier time points (1 Day after injury) to show the time course of change for this reflex.

We have modified the Results section and Figures 1 and 3 Figure 1to explain thoroughly what polysynaptic reflexes were tested. Specifically, in the figures and manuscript, we are referring to responses that had latencies between 3.5 and 12 ms which were evoked by low threshold and low frequency stimulation of a nerve whilst recording from a heteronymous nerve. We elaborated on which afferents are likely activated at the stimulation strengths we used, namely these include group Ia, Ib and II muscle afferents, cutaneous (mechanoreceptive) afferents in the group I range.

We have striven to make the descriptions of the reflex pathways more accurate and comprehensible in the Results sections and figures including their descriptions. We have provided new diagrams which we hope illustrate the reflex pathways better. We provided a new diagram for the H-reflex (Figure 1J and “Figure 1—figure supplement 3”). We have included graphs which depict the post-activation depression / frequency-dependent depression of the H-wave for baseline, Day 1, Week 2 and Week 10 time-points. In the Results section and in “Figure 1—figure supplement 3F”, we describe how we analysed the depression of the H-wave: "We measured the area under the stimulus-response curve from inter-stimulus intervals of 10 seconds to 0.1 seconds (Figure 1—figure supplement 3F) for each rat at each time point separately and then plotted the group mean values (Figure 1L)".

Furthermore, we included new diagrams (replacing old ones) for the experimental set-up of polysynaptic reflex assessments (Figure 1M, N, R and “Figure 1—figure supplement 4”), which clearly depict the nerves that were stimulated and recorded from without including any hypothesis about the complex spinal circuitry that is involved in these polysynaptic reflexes.

*2) Having introduced the model and its changes, the NT3 effects should be described in a separate text and figure with the essential reflexes and motor behavior summarized. In this way the authors can avoid jumping back and forth in the description of the model, the electrophysiological findings, and the behavioral characterizations and how these parameters are relevant for spasticity and the treatment with NT3 injections.*

We agree. We have described the effects of NT3 in separate text (the second part of the Results section) and separate figures (i.e., Figures 2-5 Figure 2including their figure supplements) with the essential reflexes and behaviours summarized there.

*3) The definition of spasticity should be clear. However, the discussion of what spasticity really is does not seem very appropriate. There is no right or wrong answer to that (and no "controversy", even if there may be strong opinions). Clearly definitions have to be somewhat different in the clinic and in animal experiments, even if one tries to find animal models that are as close to the clinical situation as possible. It should be clearly stated that selective pyramidal lesion is a rare clinical event and stroke or spinal cord injury related spasticity is much more common.*

We have revised our definition of spasticity (found in the first paragraph in the Introduction) and now provide a single clear definition of spasticity, which is also consistent with the animal behaviour we observe. We have revised the Discussion and Methods sections with regards to definitions of spasticity. We agree that definitions of spasticity may have to be different in the clinic and in animal experiments. In the Discussion section, we removed the section using the word “controversy” and acknowledge the range of spinal cord injury animal models that display spasticity that is similar to the human condition. In our Methods, we provide “behavioural operational definitions” for each of the signs of spasticity which we observed in the open field.

Furthermore, we now state clearly that selective pyramidal lesion is a rare clinical event (subsection “Bilateral corticospinal tract lesioning is sufficient to cause spasticity”, first paragraph) and that stroke or spinal cord injury-related spasticity is very common: we now present estimates for the prevalence of spasticity amongst the spinal cord injury community in our Introduction (first paragraph).

*4) There are generally a number of overstatements that NT3 treatment completely alleviates spasticity, which is not the case. This should be rectified. Also it should be noted that other treatments have been introduced in addition to baclofen.*

This was unintentional. We have modified the manuscript at the relevant sections to avoid overstating our claims about NT3 treatment.

We now acknowledge other treatments in addition to baclofen including tizanidine and botulinum toxin in the Introduction.

*5) Synaptic analysis is non-trivial and the authors state that they counted a synapse if vGAT and vGlut1 were in 'very close apposition'. What exactly do they mean by this – how close does this need to be to be classed as a synapse and without deconvolution do they think they are resolving individual synapses?*

We agree that an important distinction exists between identification of individual synapses (e.g., by electron microscopy or high resolution microscopy with deconvolution) and identification of bouton-like structures (e.g., by confocal fluorescence microscopy).

We have modified our manuscript to ensure we describe our Methods and Results more appropriately and more fully. Throughout the manuscript, we now refer to “boutons” instead of “synapses”. Instead of “very close apposition” we now use the term “in close proximity” and we define this term in Methods (Image Analysis: Analysis of synaptic boutons by confocal microscopy).

*6) How were the 5-HT changes assessed (it is not obvious from the figure that the changes are representative). Needs discussion and comments on how the quantification was done.*

In the Methods section, we now describe more clearly how the 5-HT changes were assessed (Image Analysis: Analysis of serotonin immunohistochemistry) which is an adaptation of a validated method for quantification of serotonergic fibres that has been previously published (Grider, Chen, and Shine 2006).

We have created new a figure which shows a representative image from each group rather than showing compressed heat-maps of the grey matter for each group. These images now better illustrate the mean changes depicted in the graphs.

*7) The NT3 treatment is not easy to understand. What does it do, and how can it act non-retrogradely – is it because the main effect is systemic? Otherwise it is hard to understand that so many symptoms can be affected by a single or few muscle injections. The authors should explain the large tendency in NT3 increase in triceps after virus injection in flexor muscles, and the recovery in radial MNs Figure 4D and in TrkC neurons in Figure 7B and generally discuss the systemic effects.*

We have rewritten the manuscript to make the NT3 treatment easier to understand. The Results section includes a new subsection that is dedicated to the data related to synthesis and transport of NT3 and Figure 2 presents our experimental data relating to this.

In the Methods section, we list the forelimb muscles that received treatment, i.e. how many injections and which volumes were made into which muscle groups. We administered an AAV1 which carries the human Neurotrophin-3. Our data show that Neurotrophin-3 protein is transported from forelimb flexor muscles to ipsilateral cervical DRG and to the dorsal horn of the cervical spinal cord as well as to ventral horn motor neurons. In new subsections in the Discussion ("Did Neurotrophin-3 reduce spasticity via retrograde signaling?" and "Mechanisms of recovery with neurotrophin-3") we now explain what NT3 treatment does and how it can act: we propose that the dominant mechanism involves retrograde transport although we note that we cannot exclude systemic effects.

As requested, we now discuss in the Results and Discussion sections that there was a trend (albeit not statistically significant) towards increased Neurotrophin-3 levels in the ipsilateral (and not contralateral) *triceps brachii* and we comment that this may have resulted from AAV spread from during injection of nearby muscles (e.g., *biceps brachii*) or transport of protein in the bloodstream. We now present the data for both ipsilateral and contralateral triceps and biceps in Figure 2 and its supplements.

As requested, we also now discuss the recovery of KCC2 in the radial MNs (former Figure 4D) in the Discussion section entitled "Mechanisms of recovery". We discuss the evidence that KCC2 levels might have been modified either by direct retrograde transport of neurotrophin-3 in motor axons (which might mean even the non-significant trend of increased neurotrophin-3 protein in the *triceps brachii* was sufficient to affect its radial motor neurons), or, secondarily to changes in afferent or supraspinal fiber organization.

With regard to the potential for systematic effects of Neurotrophin-3, we wish to disclose to the reviewers that my team has shown that one month of subcutaneous infusion of recombinant Neurotrophin-3 can normalise an H reflex and promote sensorimotor recovery in elderly rats when treatment is initiated 24 hours after large cortical stroke (Duricki et al., unpublished work). However, a very high dose of Neurotrophin-3 was infused subcutaneously (47 μg/kg/day) to achieve this result and we have evidence that lower doses are not effective when given systemically. Accordingly, we think it is likely that the majority of results reported in the present manuscript are due to retrograde transport of Neurotrophin-3 rather than systemic effects of Neurotrophin-3 in the blood stream. We are happy to provide this additional manuscript to reviewers if requested.

*8) The explanation of the NT3 increase in DRG by its retrograde transport is not supported by the data. Retrograde transport of the active signaling from the nerve endings to the soma is the basis of the long-range actions of the neurotrophins.*

We agree that retrograde transport of an active signalling complex from the nerve endings to the soma is the basis of the long-range actions of endogenous neurotrophins, as shown by Silvia Arber’s group for gene expression by postnatal mouse proprioceptors, for example (Lee et al. 2012). In the manuscript we also now cite several papers that also found retrograde transport of Neurotrophin-3 from muscle to DRG neurons (DiStefano et al. 1992, Helgren et al. 1997).

As described above (Essential Revisions: point 7), we have rewritten the Results section to explain better our evidence that Neurotrophin-3 is transported retrogradely from forelimb flexor muscles to ipsilateral cervical DRG and to the dorsal horn as well as motor neurons in the ventral horn of the cervical spinal cord. Specifically, our ELISA measurements of homogenates of DRG show that Neurotrophin-3 protein is elevated in ipsilateral DRGs of AAV1-NT3 rats relative to contralateral DRGs and to AAV1-GFP rats.

*9) The PCA analysis does not provide much information, nor does it strengthen the data because it relies on author observations/analysis that have been arbitrarily chosen. Should be deleted.*

As requested, we have removed the PCA analysis from the manuscript.

*10) The RNA-seq data should be presented in a comprehensive way with the genes in the main figure. However, since RNAseq has been performed on entire DRGs, it is hard to draw any conclusions from the data. Other neuron types express TrkC in the DRG and are likely to respond to NT3 overexpression. It is therefore not easy to understand the focus on sema 4. It is interesting but there is not proof or even slight indication that this molecule is involved in any of the changes. To show this would require a KO. We think it unreasonable to ask for this for this study, but we think the data are too preliminary at this point to include as an explanation and propose that the entire RNA-seq story be deleted from the paper and saved for a separate publication where the functional role of individual molecules can be addressed.*

As requested, we have removed from the manuscript the RNAseq data and the accompanying hypothesis and data about Sema4C. Thank you for the very reasonable position that you took regarding the need for KO work. I plan to apply for funding to use conditional KOs to evaluate the mechanisms by which Neurotrophin-3 promotes these changes and I will seek to publish this data in a separate publication in the future.

*11) Discussion: The comments on Lawrence and Kuypers' experiments/films are important since the present model is a bilateral lesion of the pyramidal tract! When it comes to the mechanisms (subsection “Underlying mechanisms and treatment choice”) it is not correct that most GABAB receptors are responsible for presynaptic inhibition of Ia afferents. It is also striking that the KCC2 is the only mechanism proposed for cellular mechanisms for the hyperexcitability of motoneurons. What about the persistent inward currents that were first described by Schwindt and Crill, then by Hounsgaard, Hultborn, Kiehn et al., and Dave Bennett et al. (the 2 latter groups also proving that they are increased following chronic spinal cord lesion, and certainly contribute to spasticity)? This is complemented by experiments by several other groups in Seattle, Chicago and Halifax. This paper is not addressing this question, but a couple of sentences in the Discussion should be a minimum. Finally, how can removal of the CST lead to change in competition onto motor neurons. CSTs do not reach MNs in rats?*

Thank you for these corrections and suggestions for additions; we have modified our Introduction and Discussion sections accordingly.

We now describe the role of persistent inward currents in the hyperexcitability of motor neurons controlling the hindlimbs and tail of various species including rats and work by the authors that you mention above. We also mention that changes in descending monoaminergic pathways can enhance persistent inward currents (Result section). Our model involved axotomy of non-monoaminergic pathways; this also induced changes in the pattern of monoaminergic (serotonergic) arbors in the cervical spinal cord. In the future it would be interesting to determine whether bilateral pyramidotomy induces persistent inward currents that might contribute to the moderate spasticity that we observed.

We agree that corticospinal tract neurons do not or rarely reach MNs directly in rats. At present we do not know why removal of the CST leads to an increase in vGluT1 positive boutons on motor neurons. Prof. Jack Martin’s group has also observed a similar phenomenon in the cervical spinal cord (ten days after unilateral pyramidotomy) and we now cite their work (Tan et al. 2012). We also now discuss the possibility that bilateral CST axotomy leads to anterograde transneuronal loss of synapses from motor neurons in the Discussion section "Importance of proprioceptive feedback". Complete bilateral ablation of the CST will cause considerable loss of input to spinal interneurons and it is possible that this loss of descending activity leads to loss of synapses from these interneurons onto motor neurons. In turn a signal from spinal cord motor neurons might lead to synthesis and secretion of factors known to promote sprouting and synapse formation. Indeed, evidence from other groups shows that pyramidotomy leads to changes in the expression profile in the cervical spinal cord of genes known to play roles in axon sprouting and synapse formation (Bareyre and Schwab 2003). This might cause sprouting and synapse formation of vGluT1 expressing afferents onto motor neurons, which we observed in our study.

*12) Figure 8 should be deleted since it there is not support for only retrograde NT3 transport as the mechanism.*

As we argue above, the majority of our data are consistent with a predominant effect of NT3 via retrograde transport. Notwithstanding this, we accept that we cannot exclude some systematic effects of NT3.

Accordingly, we have modified our summary figure and its figure legend to show that NT3 may be acting retrogradely as well as systemically.

*13) Title: The title needs to be changed to reflect the revision. In particular it needs to be stated what the CNS injury is and what reflexes that are normalized.*

We have changed the title to reflect the revisions. We now state the type of CNS injury (bilateral corticospinal tract injury) and that low threshold spinal reflexes were normalized. In the Abstract we now state clearly that these also included the H reflex to a hand muscle and low threshold polysynaptic spinal reflexes.